# DataMIL: Selecting Data for Robot Imitation Learning with Datamodels

**Shivin Dass**[*]
UT Austin

**Alaa Khaddaj**[*]
MIT

**Logan Engstrom**
MIT

**Aleksander Mądry**
MIT

**Andrew Ilyas**[†]
Stanford University

**Roberto Martín-Martín**[†]
UT Austin, Amazon

## Abstract

Recently, the robotics community has amassed ever larger and more diverse datasets to train generalist policies. However, while these policies achieve strong mean performance across a variety of tasks, they often underperform on individual, specialized tasks and require further tuning on newly acquired task-specific data. Combining task-specific data with carefully curated subsets of large prior datasets via co-training *can* produce better specialized policies, but selecting data naively may actually harm downstream performance. To address this, we introduce DataMIL, a data selection framework built on the datamodels paradigm that reasons about data selection in an end-to-end manner, using the policy itself to identify which data points will most improve performance. Unlike standard practices that filter data using human notions of quality (e.g., based on semantic or visual similarity), DataMIL directly optimizes data selection for task success, allowing us to select data that improves the policy while dropping data that degrade it. To avoid performing expensive rollouts in the environment during selection, we introduce a surrogate loss function on task-specific data, allowing us to use DataMIL in the real world without degrading performance. We validate our approach on 60+ simulation and real-world manipulation tasks, notably showing successful data selection from the largest open collections of robot datasets (OXE); demonstrating consistent gains in success rates over prior works. Our results underscore the importance of end-to-end, performance-aware data selection for unlocking the potential of large prior datasets in robotics. More information at our [website](#).

## 1 Introduction

Recently we have witnessed a revolution in robot learning: inspired by the successes of large-scale language and vision models, the robotics community has begun training large foundation policies (Brohan et al., 2023; Black et al., 2024; Bjorck et al., 2025) by amassing large diverse robotic datasets comprising of a variety of tasks, scenes, and robots (O'Neill et al., 2024; Walke et al., 2023; Fang et al., 2024). The resulting generalist policies achieve a strong mean performance across tasks and environments, but often underperform on individual tasks (Kim et al., 2024; 2025), highlighting a gap between generalization and task-specific competence. To bridge this gap, researchers have explored a post-training paradigm (Black et al., 2024), where pre-trained foundation models are fine-tuned (Black et al., 2024; Kim et al., 2025) for specific tasks, though this process demands a considerable number of newly acquired task-specific demonstrations. As datasets grow increasingly large and diverse, a natural question arises: *how can we identify and select data from within existing datasets to boost task performance?*

Selecting data to train a high-performing model is a complex endeavor. Naively, it would require testing **each subset** of the data by retraining and evaluating the performance of the trained model. This is expensive for any sizable dataset, becoming infeasible in robotics, where evaluation involves

---

[*]: denotes equal contribution. [†]: denotes equal advising
Correspondence to: `sdass@utexas.edu`

policy rollouts in the real world—a time-consuming and often dangerous procedure. Prior data selection methods in robotics remove the dependency on policy rollouts by selecting data based on heuristics, i.e., assuming that the most useful data is the most similar in language description Zha et al. (2024), visually (Memmel et al., 2024), in motion (Lin et al., 2024), or in state-action pairs (Du et al., 2023) to a small number of task-specific demonstrations. While intuitive (and effective in some cases), these heuristics often make several assumptions and fail to consider the real impact of a datapoint on policy performance (see Fig. 1 (left)).

In other fields such as natural language processing (NLP) and computer vision (CV), researchers have developed an efficient framework for data selection based on model performance: **datamodels** (Ilyas et al., 2022). Training a datamodel is a meta-process in which the original learning algorithm and model are viewed as a "black box" that consumes data, and the goal is to *train an estimator of the black box's behavior as a function of the input data*. By avoiding selecting based on heuristics, datamodels stay close to the true optimization objective (trained model performance) while critically reducing the amount of training and evaluation steps necessary with the original learning algorithm to an acceptable level for NLP and CV. However, these evaluations are still infeasible for policy learning due to the need for real-world rollouts, impeding their application to robotics.

In this work, we introduce **DataMIL** (Datamodels for Imitation Learning), a method that extends the success of datamodels to robotics by addressing the unique challenges of data-driven data selection. DataMIL trains a data-quality estimator using a tractable surrogate objective. We demonstrate empirically that this objective retains sufficient correlations to the true objective (trained policy performance), allowing us to train datamodels that predict data influence without requiring expensive real-world rollouts. Moreover, thanks to a process that remains closer to the true objective and avoids assumptions about the characteristics of the useful data, DataMIL selects and curates datasets for hard cases (e.g., different embodiments, multi-task settings) where prior heuristic-based methods are unsuitable. Across 50 tasks in MetaWorld (Yu et al., 2020), we show a 10% performance boost compared to the state-of-the-art baselines. We then show how datamodels can be estimated efficiently for larger policies such as Octo (Octo Model Team et al., 2024) using improved datamodel estimators based on metagradients (Engstrom et al., 2025) and show task-specific dataset curation in ten tasks from the LIBERO benchmark (Liu et al., 2023). Finally, we effectively select data from OXE (O'Neill et al., 2024), one of the largest open collections of robot datasets, demonstrating DataMIL's efficacy in selecting datasets for new tasks and embodiments on real hardware.

## 2 Related Work

**Data curation for robot learning.** Recent advances in robotics have leveraged ever-larger demonstration collections—both in simulation (Mandlekar et al., 2021; 2023; Sharma et al., 2018), and on real hardware (Walke et al., 2023; Khazatsky et al., 2024; Mandlekar et al., 2018)—to train generalist policies capable of tackling diverse tasks (Brohan et al., 2023; Kim et al., 2024; Black et al., 2024; Bjorck et al., 2025). However, the sheer scale and heterogeneity of these datasets (varying robots, scenes, and objectives) has motivated a body of work on data curation. For generalist training, methods like Re-Mix (Hejna et al., 2024) use DoReMi (Xie et al., 2023) style optimization to learn optimal mixtures of data domains for improving model training, while others identify "high-quality" trajectories via mutual information criteria (Hejna et al., 2025) or by scoring samples with policy rollouts (Chen et al., 2025).

Beyond generalist policy training, many studies have focused on task-specific dataset selection: given a handful of target demonstrations, one can sub-sample large datasets based on visual similarity (Memmel et al., 2024), motion cues (Xu et al., 2022), or state–action closeness (Du et al., 2023). Recent work (Kumar et al.), aims at fusing datasets aggregated via different similarity metrics by scoring each data subset and re-sampling them according to a weight proportional to this score. While these approaches capture human notions of quality, they remain agnostic to each sample's actual impact on downstream policy performance. Concurrent to our work, CUPID (Agia et al., 2025) focuses on single task curation by using a policy-gradient influence measure estimated via online rollouts. In contrast, DataMIL demonstrates selection from large heterogeneous datasets, scoring datapoints entirely in an offline manner. This allows us to estimate each datapoint's contribution to final task success and curate training sets in an end-to-end, performance-aware fashion.

Figure 1: **Data selection with datamodels.** (left) Similarity-based methods select *close* samples (yellow), but these aren't always beneficial for learning. DataMIL evaluates data based on its impact on policy performance, selecting the samples that lead to policy improvement. (center) We estimate datamodels that score each sample by its influence on policy performance and select the highest-scoring samples for training. (right) DataMIL explores two datamodel estimation methods, adapted to robotics: *regression* and *metagradient-based* estimation (see Sec. 4.1).

**Datamodels and data attribution.** Our work draws from a line of work in machine learning on data attribution (Koh & Liang, 2017; Ilyas et al., 2024; Hammoudeh & Lowd, 2024; Park et al., 2023; Chang et al., 2024; Ilyas & Engstrom, 2025) and, in particular, the datamodels framework (Ilyas et al., 2022; Park et al., 2023). At a high level, this framework seeks to predict the behavior of machine learning models as a function of the data they are trained on. These ideas have been explored in the fields of CV and NLP for improving language model pre-training (Engstrom et al., 2024) and instruction tuning (Xia et al., 2024; Liu et al., 2024; Engstrom et al., 2025); for increasing worst-group robustness (Jain et al., 2024); and for removing outliers in supervised learning settings (Lin et al., 2022). Our work builds on this body of research, while—as we discuss in Sec. 4.2—also tackling the unique challenges posed by the robot learning setting.

## 3 PRELIMINARIES: DATA SELECTION FOR POLICIES AND DATAMODELS

In this section, we first provide some general background on the problem of data selection for robot learning, and then describe the datamodels framework on which our method is based.

**Policy learning.** The focus of our work is on the *imitation learning* problem. Here, our goal is to learn a policy $\pi$ that maps states $s$ to distributions over actions $a$ using a collection of training trajectories (or *demonstrations*) $\mathcal{D} = (\tau_1, \tau_2 \cdots \tau_n)$ of state-action pairs. We will define a policy learning *algorithm* $\mathcal{A}$ as a function that takes as input a dataset of demonstrations $\mathcal{D}$ and outputs an optimized policy $\pi$. We measure the performance of the policy $\pi$ using a *metric* $\mathcal{M} : \pi \to \mathbb{R}$, which is a function mapping policies to a scalar value. The most common choice of metric is *success rate*—the fraction of times that sampling actions from the policy results in the policy completing a given task—but our notation is general and can also capture other choices of metric $\mathcal{M}$.

**Data selection in robotics.** In data selection for robotics, we are given (a) a prior dataset $\mathcal{D}$ of demonstrations; (b) a fixed learning algorithm $\mathcal{A}$ (e.g., stochastic gradient descent on an imitation learning objective); and (c) a target metric $\mathcal{M}$ that we will use to measure policy performance. Our goal is to select a subset of the data $\mathcal{D}' \subset \mathcal{D}$ such that applying the algorithm $\mathcal{A}$ to the subset $\mathcal{D}'$ yields a policy $\pi$ that maximizes the target $\mathcal{M}$. Formally, we aim to find

$$\underset{\mathcal{D}' \subset \mathcal{D}}{\arg \max} \, \mathcal{M}(\mathcal{A}(\mathcal{D}')). \tag{1}$$

Solving this optimization problem is challenging since algorithm $\mathcal{A}$ itself is expensive to compute (it involves training a policy on $\mathcal{D}'$). Thus, exhaustive search over all subsets of $\mathcal{D}$ is infeasible.

**Datamodels.** In this work, we leverage data attribution (Koh & Liang, 2017; Ilyas et al., 2024; Park et al., 2023; Bae et al., 2024) (and specifically, the datamodels framework (Ilyas et al., 2022)) to tackle the data selection problem in Eq. 1. The key idea behind datamodels is to directly approximate

the target metric $\mathcal{M}(\mathcal{A}(\mathcal{D}'))$ as a function of the data $\mathcal{D}'$, allowing us to answer questions like "what would the performance of the policy be if we trained on this subset of the data?" without actually training the policy. More precisely, a datamodel is a model $\hat{f} : 2^{\mathcal{D}} \to \mathbb{R}$ that takes as input a subset of the data $\mathcal{D}' \subset \mathcal{D}$ and outputs an estimate of $\mathcal{M}(\mathcal{A}(\mathcal{D}'))$.

**Informal Definition 1 (Datamodeling problem)** *Given a prior dataset $\mathcal{D}$ and a learning algorithm $\mathcal{A}$, the* datamodeling problem *is the problem of predicting the behavior of a model trained on a subdataset $\mathcal{D}' \subset \mathcal{D}$ without actually training a model.*

If we had such an approximation in hand (assuming for now that we can compute one), we would approach the data selection problem in Eq. 1 by solving the following optimization problem:

$$\arg\max_{\mathcal{D}' \subset \mathcal{D}} \hat{f}(\mathcal{D}'). \tag{2}$$

This problem is much more tractable than the original problem in Eq. 1 because the datamodel $\hat{f}$ is typically much cheaper to compute than the learning algorithm and target metric $\mathcal{M}(\mathcal{A}(\cdot))$. For example, in supervised learning, recent works have shown that even *linear* datamodels—functions $\hat{f}$ that decompose additively in terms of their inputs $\mathcal{D}$—can be accurate predictors of model performance (Ilyas et al., 2022; Lin et al., 2022; Park et al., 2023; Bae et al., 2024; Chang et al., 2024). Intuitively, this means that we can assign a scalar value to each data point in $\mathcal{D}$ that indicates how much it contributes to the performance of the learning algorithm $\mathcal{A}$.

Since leveraging an accurate datamodel $\hat{f}$ for data selection is straightforward, the main challenge in our work is *constructing* such a datamodel. That is, we need to find a way to build a function $\hat{f}$ that can predict, with nontrivial accuracy, the performance of the learning algorithm $\mathcal{A}$ on any given subset of the data $\mathcal{D}'$ *without actually training a policy on that subset*. In the next section, we describe our methods for constructing such a datamodel in the policy learning setting.

## 4    DataMIL: Datamodels for Robot Imitation Learning

An overview of our training and selection methodology is shown in Fig. 1. Below, we provide the details of each component, beginning with the estimation of datamodels (Sec. 4.1), adapting them to robotics (Sec. 4.2) and selecting data using datamodels to train policies (Sec. 4.3).

### 4.1    Estimating Datamodels

In our work, we consider two ways of estimating datamodels: the *regression* method (Ilyas et al., 2022) and the *metagradient* method (Engstrom et al., 2025). Both estimators approximate the outcome of model training *linearly*, in the sense that, for any training subset $\mathcal{D}' \subset \mathcal{D}$, the datamodel prediction $\hat{f}(\mathcal{D}')$ takes the form,

$$\hat{f}(\mathcal{D}') = \sum_{z_i \in \mathcal{D}'} \tau(z_i).$$

Intuitively, $\tau(z_i)$ captures the importance of the training example $z_i$ to the target metric $\mathcal{M}(\mathcal{A}(\mathcal{D}'))$ (more precisely, $\tau(z_i)$ is the *additive effect* of $z_i$ on the target metric). Linear datamodels are convenient in the context of data selection, since they allow us to solve Eq. 1 by simply selecting the training examples with the highest scores $\tau(z_i)$. Both estimators below take this form, differing only in how they compute $\tau(z_i)$.

**Regression estimator.** The regression estimator is a straightforward but expensive way to estimate datamodels—it involves *precomputing* the scores $\tau(z_i)$ for each training example $z_i$ in the broader training set $\mathcal{D}$. Concretely, we first sample $m$ random subsets of the prior dataset $\mathcal{D}_j \subset \mathcal{D}$; for each of these datasets, we train a policy $\mathcal{A}(\mathcal{D}_j)$, and evaluate the target metric $\mathcal{M}$. In our setting, evaluating $\mathcal{M}$ means rolling out the policy several times and computing the success rate. This gives us $\mathcal{M}_j = \mathcal{M}(\mathcal{A}(\mathcal{D}_j))$ for each subset $\mathcal{D}_j$, creating a dataset of $m$ data-performance pairs $(\mathcal{D}_j, \mathcal{M}_j)$, for estimating datamodels. We compute the scores $\tau(z_i)$ for all $n$ training examples by solving the

following minimization problem:

$$\{\tau(z_1), \ldots, \tau(z_n)\} := \arg \min_{\boldsymbol{\tau} \in \mathbb{R}^n} \sum_{j=1}^{m} \left( \sum_{j:z_i \in D_j} \tau_i - \mathcal{M}(\mathcal{A}(D_j)) \right)^2. \tag{3}$$

Above, observe that the sum $\sum_{j:z_j \in D_i} \tau_j$ is precisely the datamodel prediction of $\mathcal{M}(\mathcal{A}(D_i))$ when setting $\tau(z_j) = \tau_j$. Thus, Eq. 3 corresponds exactly to linearly regressing the target metric onto the presence of each training point $z_i$. More details are provided in App. B.2

**Metagradient-based estimator.** The regression-based estimator requires training a large number of models, which quickly becomes computationally prohibitive for complex models such as the ones used for modern visuomotor policies. A method to alleviate this challenge is to use a metagradient-based estimator. To that end, we parameterize the dataset by a vector $\mathbf{w} \in [0,1]^n$, where $w_i$ specifies the weight of the $i$-th training sample ($i$ is included in the dataset if $w_i = 1$ and excluded if $w_i = 0$). Treating these weights as meta-parameters, we write a first-order Taylor expansion to our objective $\mathcal{M}(\mathcal{A}(\mathbf{w}))$ from Eq. 1 as,

$$\mathcal{M}(\mathcal{A}(\mathbf{w})) \approx \mathcal{M}(\mathcal{A}(\mathbf{w}_0)) + \nabla_{\mathbf{w}} \mathcal{M}(\mathcal{A}(\mathbf{w}_0))^\top (\mathbf{w} - \mathbf{w}_0), \tag{4}$$

where $\mathbf{w}_0$ is the all-ones vector, corresponding to training on the entire dataset. The gradient $\mathcal{I} = \nabla_{\mathbf{w}} \mathcal{M}(\mathcal{A}(\mathbf{w}_0))$, known as the influence function (Hampel, 1974; Koh & Liang, 2017), captures how changes in data weights affect model performance.

Naïvely computing $\mathcal{I}$ requires differentiating through the training process $\mathcal{A}(\mathbf{w})$—i.e., unrolling policy optimization with respect to the meta-parameters $\mathbf{w}$. This computation, known as the *metagradient*, has traditionally been notoriously difficult, and consequently most prior works in data attribution have focused on approximating $\mathcal{I}$ (Koh & Liang, 2017; Park et al., 2023; Bae et al., 2024). Recent advances, however, demonstrate that metagradients can be computed *exactly* and *efficiently* by leveraging the iterative structure of standard optimization algorithms such as stochastic gradient descent (SGD) (Engstrom et al., 2025; Calian et al., 2025). Specifically, Engstrom et al. (2025) exploit step-wise auto-differentiation combined with efficient data structures to keep the memory costs practical. The resulting influence scores $\mathcal{I}_i$ quantify how positively or negatively the $i$-th training sample affects performance, and can be directly used as datamodel coefficients. Ilyas & Engstrom (2025) shows that indeed, doing this reparameterization and Taylor expansion with the metagradient actually does give you very good estimates of model performance. A detailed description and pseudocode of the metagradient-based estimator is provided in App. B.3.

## 4.2 Adapting Datamodels for Robotics

While datamodels have been applied to language modeling (Park et al., 2023; Bae et al., 2024) and computer vision (Koh & Liang, 2017; Ilyas et al., 2022) tasks, there are some unique challenges that we face when applying them to robotics. Below, we describe how DataMIL extends the datamodel framework to handle robotic datasets efficiently.

**Estimating datamodels without rollouts.** In principle, the ideal target metric $\mathcal{M}$ is the policy's true success rate under environment rollouts. However, real-world rollouts are expensive, and using them directly renders the objective non-differentiable, preventing the use of metagradient-based estimators that exploit differentiability of the evaluation metric. To overcome these limitations, we introduce a proxy metric $\widehat{\mathcal{M}}$ that (1) requires no additional rollouts and (2) is fully differentiable. Concretely, given a small held-out demonstration set $\mathcal{D}_{target}$ for the target task, we define:

$$\widehat{\mathcal{M}}(\pi, \mathcal{D}_{target}) = \frac{1}{|\mathcal{D}_{target}|} \sum_{(s,a) \in \mathcal{D}_{target}} -\mathcal{L}_{BC}(\pi(s), a) \tag{5}$$

Where $\mathcal{L}_{BC}(\pi(s), a)$ defines the policy loss on a training example $(s, a)$ (see App. D for the exact objective for different policy classes). Hence, the true target metric $\mathcal{M}$ can be substituted by the proxy metric $\widehat{\mathcal{M}}$ in our original optimization (Eq.1), resulting in a more tractable and end-end differentiable objective for applying datamodels to robotic settings.

A natural question is whether $\widehat{\mathcal{M}}$ is a sufficiently good approximation of the true target to enable data selection. We study this question using the `pick-place-wall` task from Meta-World (Yu et al., 2020) (results on more tasks are provided in App. C.1), where we consider three different datamodel estimation techniques:

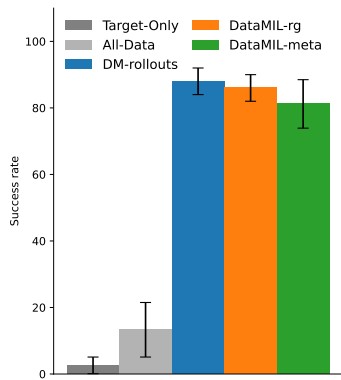

i) *DM-rollouts*: using regression-based estimator to estimate a datamodel for the "true" target $\mathcal{M}$ (success rate via rollouts);

ii) *DataMIL-rg*: using regression-based estimator to estimate a datamodel for proxy target $\widehat{\mathcal{M}}$ (loss on a heldout set);

iii) *DataMIL-meta*: using metagradient-based estimator to estimate a datamodel for proxy target $\widehat{\mathcal{M}}$.

Figure 2: Comparing true rollout success ($\mathcal{M}$) vs. proxy metric ($\widehat{\mathcal{M}}$)

We use each of these datamodels to select the top 10% of samples (as ranked by their estimated coefficient $\tau(z_i)$) from a multi-task prior dataset consisting of a mix of expert and suboptimal demonstrations (see Sec. 5 for details). We then measure the *true success rate* of a policy trained on the selected samples, and visualize the results in Fig. 2. Our results show that (a) selecting data for the proxy metric $\widehat{\mathcal{M}}$ incurs only a marginal drop in final success; (b) applying the metagradient-based estimator incurs another small drop in success but are significantly faster to train. Moreover, policies trained on selected data vastly outperform baselines trained on the entire dataset or $\mathcal{D}_{target}$ alone, achieving $7\times$ higher success than the all-data policy, while target-only policy fails almost entirely.

These results demonstrate that (a) while validation loss is often a noisy predictor of performance in robotics, it still provides a useful signal for data selection, and our proxy objective can effectively replace expensive rollouts, and (b) data curation is critical: naively using all data or only target examples yields suboptimal policies, whereas curated datasets substantially boost task success.

**Clustering training examples.** Datamodels measure the influence of each *training example* on overall policy performance. In robotics, a training example may range from a single state–action pair, to sequences of state–action pairs (for sequential policies). Estimating influence at the individual training example level is often noisy: because each training example is seen only a few times during training, its individual effect on policy performance is often marginal and difficult to measure precisely. To reduce this variance, prior works clusters samples by class or task and then evaluates cluster-level influence instead of individual examples (Jain et al., 2023; Ley et al., 2024).

This is particularly suitable for robotics since data naturally clusters into state–action sequences, so we can group samples at different temporal scales—e.g. sub-trajectories (Memmel et al., 2024), tasks (Zha et al., 2024), or entire domains (Hejna et al., 2024). Fine-grained clusters offer precise selection but suffer from high noise, while coarse clusters yield more stable influence estimates at the cost of detail. We found that the optimal clustering strategy is a function of the dataset size given a fixed compute budget: the larger the dataset, the fewer times each sample is seen during training, leading to noisier individual influence estimates. On mid-sized datasets like LIBERO, sub-trajectory clustering provides a balance between granularity and noise (see App. C.4), whereas on large-scale datasets such as OXE, aggregating trajectory-level influence yields more reliable influence estimates.

**Reducing distribution shift.** Distribution shift is a particularly pronounced challenge in robotics: slight changes in lighting, camera pose, or robot dynamics can dramatically alter the data distribution. This issue becomes more severe when selecting data from large, heterogeneous datasets comprising of a variety of different embodiments, scenes, and tasks. Since datamodels rely on training policies over the prior data to estimate their impact on target task performance, excessive shift between the training and target distributions can lead to poor datamodel estimates. To mitigate this, we include a small fraction of target task data during datamodel estimation to better align the policy's learning with the target domain. Specifically, we split $\mathcal{D}_{target}$ in half, include one half alongside the prior data for datamodel estimation, and reserve the other half purely for evaluating the proxy objective. We apply this technique only in real-world settings (i.e., OXE) since in simulation (i.e., MetaWorld, LIBERO) the target data typically comes from a similar distribution as the prior data.

### 4.3 DATA SELECTION AND POLICY TRAINING

By applying our proposed modifications in Sec. 4.2 to the datamodel estimators described in Sec. 4.1, we obtain a per-cluster attribution score on policy performance. While these attribution scores can be used in a variety of different ways, in this work we use them to curate a training subset: we select the top $x\%$ of prior examples with the highest positive influence to form $\mathcal{D}_{sel}$. We then train the downstream policy $\pi$ via behavior cloning on $\mathcal{D}_{sel}$ and $\mathcal{D}_{target}$ using a co-training recipe (Lin et al., 2024; Maddukuri et al., 2025; Khazatsky et al., 2024; Nasiriany et al., 2022): at each training step, we sample from $\mathcal{D}_{target}$ with probability $\alpha$ and from $\mathcal{D}_{sel}$ with probability $1-\alpha$.

In summary, DataMIL leverages datamodels (Ilyas et al., 2022) to estimate how individual training examples effect a policy's performance on a given task. We propose key modifications that make this estimation tractable and robust in robotics settings—reducing noise, avoiding expensive rollouts, and minimizing distribution shift. Given a prior dataset $\mathcal{D}$ and target dataset $\mathcal{D}_{target}$, we **(1) Cluster** the prior data $\mathcal{D}$ into trajectories or sub-trajectories, **(2)** Estimate influence scores using our proposed **proxy metric**, with the *regression* or *metagradient* datamodel estimators, **(3)** Select the top ranked clusters creating $\mathcal{D}_{sel}$, and **(4)** Train a final policy co-trained on the target and selected data.

## 5 EXPERIMENTS

**Datasets.** We test DataMIL on two widely used multi-task simulation benchmarks: (1) Meta-World (Yu et al., 2020), contains a suite of 50 distinct robot manipulation tasks, and (2) LIBERO benchmark (Liu et al., 2023), consists of 100 tasks with diverse objects, layouts and scenes. In the real world we test using the Open-X Embodiment (OXE) datasets (O'Neill et al., 2024) – an aggregation of diverse robotic datasets collected across various robots and labs around the world.

**Training and Evaluation Details.** We use the language-conditioned Octo (Octo Model Team et al., 2024) model in LIBERO and OXE settings, initializing the model with a pretrained checkpoint provided by the authors to speed up training. For MetaWorld, we use the environment state as policy input, and hence use a simpler MLP based policy with a Gaussian action head, and study both goal-conditioned and no-conditioning settings. Results for the latter can be found in App. C.2.

**Baselines.** We compare DataMIL to the following prior works: BehaviorRetreival **(BR)** (Du et al., 2023) trains a VAE on state-action pairs and uses similarity with the target data in the latent space to retrieve single state-action pairs; FlowRetrieval **(Flow)** (Lin et al., 2024) uses a similar approach but trains the VAE on the flow features of the images computed using GMFlow (Xu et al., 2022); **STRAP** (Memmel et al., 2024) uses features from a DinoV2 (Oquab et al., 2023) model and uses dynamic time-warping to retrieve similar sub-trajectories. We also introduce a simple action retrieval **(AR)** heuristic that retrieves based on action sequence similarity. Finally, we train policies only on the target data (**Target-Only**), and co-trained with all data (**All-Data**) to measure the overall importance of data selection.

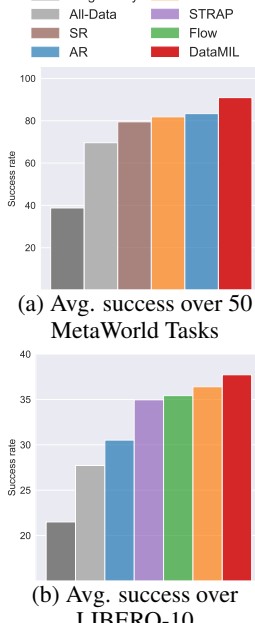

(a) Avg. success over 50 MetaWorld Tasks

(b) Avg. success over LIBERO-10

Figure 3: Performance of policy trained on selected datasets in sim. environments.

### 5.1 RESULTS

*How does data selected using DataMIL impact policy performance?* **Metaworld.** MetaWorld's 50 manipulation tasks offer a rigorous testbed for data selection. We construct our prior multi-task dataset $\mathcal{D}$ by combining (1) expert demonstrations generated by scripted policies and (2) lower-quality exploration trajectories sampled from the replay buffer of a multi-task SAC agent trained across all tasks (see App. F.1 for details). For each task, we use 5 expert demos as $\mathcal{D}_{target}$, and use the regression-based datamodel estimator to select the top 10% of samples (Sec. 4).

This setup is challenging as the selection method must both identify relevant tasks and filter noisy, suboptimal actions from the autonomous data. In Fig. 3a, we report policy performance averaged over all 50 tasks. Similarity-based baselines perform poorly: state-only (**SR**) fails to re-

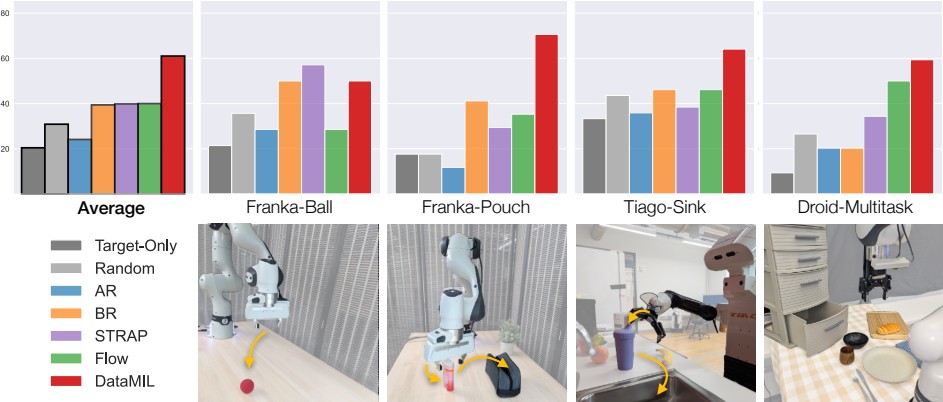

Figure 4: **Results on OXE.** We evaluate policies trained on subsets of the Open X-Embodiment dataset selected by different strategies. DataMIL consistently achieves the highest success, highlighting the need for end-end policy-aware data selection techniques. (**Droid-Multitask** shows average success; per-task results in App. C.7.)

ject poor actions, action-only (**AR**) selects irrelevant tasks with similar action distributions, and state-action retrieval (**BR**) gives equal weight to both modalities, which may not be the appropriate recipe for all tasks (App. A.1 provides a qualitative analysis). In contrast, by directly estimating each sample's influence on policy performance, DataMIL effectively identifies useful demonstrations and discards harmful samples.

*Can we scale DataMIL to larger and more complex policy classes?*

**LIBERO.** We test DataMIL on 10 long-horizon tasks from the LIBERO-10 setting, using LIBERO-90 (comprising 4500 human-teleoperated demonstrations) as the prior dataset and selecting 10% of the data (App. C.3 compares other selection %). The complex tasks and the high-dimensional RGB observations in LIBERO demand a powerful policy; we use Octo (Octo Model Team et al., 2024), a transformer-based diffusion policy. Training Octo is costly, making the regression estimator (which requires retraining across many subsets) impractical, and so we employ the metagradient datamodel estimator (Sec. 4.1).

Fig. 3b compares success rates of policies trained on the data selected by DataMIL with the baselines in each of the 10 target tasks. The relatively clean structure of LIBERO—single-view, single embodiment—makes it favorable for baselines that select via visual similarity (eg. **STRAP**, **BR** and **Flow**). However, we observe that their effectiveness varies significantly across tasks, likely due to the task-dependent suitability of each heuristic (task-wise success rates are provided in the appendix Tab. 1). In contrast, DataMIL consistently performs well across all tasks, achieving the highest average performance overall.

*Can we select data from large heterogeneous datasets in the real world?*

**Open X-Embodiment Dataset (OXE).** In the real world, we show data selection from the OXE (O'Neill et al., 2024) datasets and evaluate on four tasks on two robot embodiments shown in Figure 4 (top). This setting is particularly challenging: OXE is a heterogeneous aggregation of data from different labs, robots, camera setups, lighting conditions, and object arrangements. Further, none of our test tasks appear in OXE—we avoid matching the scene, camera pose, or objects. Instead, we aim to understand whether seemingly unrelated prior data can still yield positive transfer when curated appropriately. Our base setup uses 24 OXE datasets that were part of Octo's original training. For the **Franka-Ball** and **Franka-Pouch** tasks, we subset this to 13 and 23 datasets, respectively (denoted as OXE-13 and OXE-23). Full details on the number of target demonstrations, dataset partitions, and evaluation methodology are provided in App. F.3.

Due to the scale of OXE, we replace the **All-Data** baseline with a **Random** baseline that randomly samples the same number of datapoints as DataMIL and other methods. In Figure 4, we observe that DataMIL effectively selects relevant data even from highly heterogeneous sources. In the simpler **Franka-Ball** task, visual similarity-based baselines perform competitively. However, as the dataset grows more diverse—as in **Franka-Pouch**—these heuristics begin to break down, while DataMIL

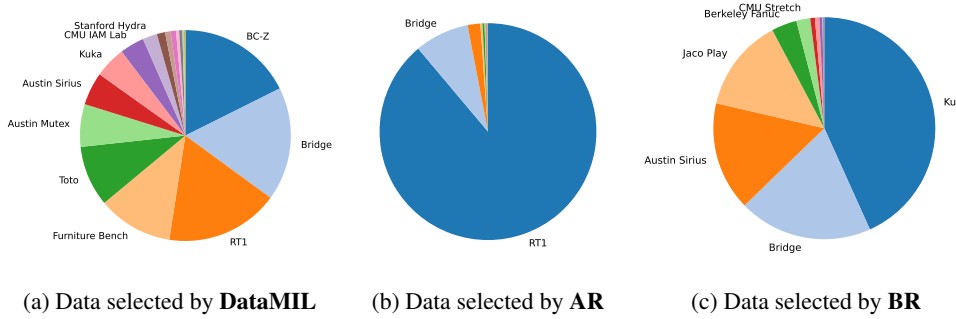

(a) Data selected by **DataMIL**    (b) Data selected by **AR**    (c) Data selected by **BR**

Figure 5: Distribution of datasets selected by different methods for **Tiago-Sink** task

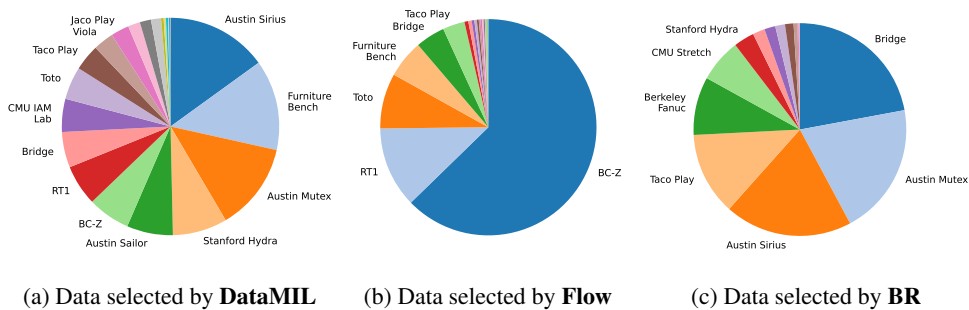

(a) Data selected by **DataMIL**    (b) Data selected by **Flow**    (c) Data selected by **BR**

Figure 6: Distribution of datasets selected by different methods for **Franka-Pouch** task

continues to identify data that improves policy performance. In the **Tiago-Sink** task, we explore a harder setting, selecting data for the Tiago (Pages et al., 2016) robot, an embodiment that never appears in the prior data. Despite this, DataMIL is able to select cross-embodiment demonstrations that improve task success. We discuss this further in Sec. 5.2. Finally, we move beyond single-task selection in the **Droid-Multitask** setting, where the target comprises three tasks: *bread in bowl*, *napkin in drawer*, and *open drawer*. This setting tests whether a single curated dataset can support multiple downstream objectives simultaneously. DataMIL consistently outperforms baselines, selecting data that improve its performance on all tasks, yielding a stronger average overall.

These results highlight that DataMIL scales to real-world robotics, handles heterogeneous datasets, and supports both single-task and multitask learning—even in settings with unseen embodiments.

## 5.2 WHAT DATA IS SELECTED BY DATAMIL?

Here we discuss and qualitatively analyse the data selected by DataMIL from the OXE dataset. More visualizations are shown in App. A.3.

**Type of embodiments selected.** DataMIL is able to select useful data for a completely new embodiment in the **Tiago-Sink** experiment. In Fig. 5a we show the highest frequency datasets selected by DataMIL and observe that even though they are visually quite different (Fig. 10c), sampled from datasets such as RT-1 (Brohan et al., 2022), BC-Z (Jang et al., 2022) and Bridge (Ebert et al., 2021; Walke et al., 2023), they still represent the essence of the target task – robots operating on a table top from an ego-perspective. For baselines, we observe that even when the target embodiment is present in the prior dataset (e.g., Franka), the selected data often comes from unrelated domains. For instance, in **Franka-Pouch** (Fig. 6), **Flow** selects data from *BC-Z* (Google Robot), and **BR** retrieves from *Bridge* (WidowX). This could be due to the baselines placing more weight on the scene/distractors when computing similarities. In contrast, the top five most frequently selected datasets by DataMIL are all sourced from Franka (Fig. 6a).

**Distribution of selected data.** In Figures 5 and 6, we show the distribution of datasets selected by DataMIL and representative baselines for the **Tiago-Sink** and **Franka-Pouch** tasks, respectively.

We find that data selected by DataMIL usually balances several different datasets, whereas most baselines select a majority of their data from a single source. For example, **AR** retrieves most of its data from *RT-1* in the **Tiago-Sink** task (Fig. 5b), while **Flow** disproportionately selects samples from *BC-Z* for **Franka-Pouch** (Fig. 6b). In contrast, DataMIL consistently selects data across a broader range of datasets in both of these cases (Fig. 5a and 6a). We hypothesize that since there is no data that exactly matches the target task, the selected data must not only be relevant but general, so as to enable positive transfer in capabilities and not make the policy overfit to a single type of domain.

**Top and bottom samples.** Analyzing the the highest and lowest ranked datapoints by DataMIL in Fig. 10 we find that they typically look similar (e.g., same embodiment or dataset). This makes sense: similar states can have very different action distributions, and while some of these actions might help reduce the policy loss on the target data, the others might lead to a large deviation, making them *harmful* for final policy learning. This aligns with data attribution works in computer vision, where harmful data looks very similar to helpful data but with different labels (Ilyas et al., 2022; Feldman & Zhang, 2020). Understanding how and why these fine-grained differences affect data selection, and ultimately, the policy performance, is an interesting direction for future work.

## 6    CONCLUSION

We present DataMIL, a data-driven method for data selection for imitation learning. DataMIL builds upon the framework of datamodels, which has been applied successfully to data selection in NLP and CV, and extends it to real-world robotic applications. Extensive experiments in simulation and real-world settings empirically support that DataMIL retrieves data to train higher-performing policies than multiple existing state-of-the-art baselines, particularly in complex scenarios. At the same time, several limitations remain. First, despite employing an efficient metagradient-based estimator, estimating datamodels still incurs computational costs several times higher than training a policy on all data (see App. G), making scalability an important challenge. Accelerating datamodel training, for instance through smaller proxy models (Khaddaj et al., 2025), is a promising direction. Second, DataMIL (and data selection techniques for robot learning more broadly) relies on a range of hyperparameters (e.g., target dataset size, clustering sizes) for which we currently lack strong intuitions. We expect future iterations and broader adoption to make these choices more principled and user-friendly. Finally, while we demonstrated selection from large prior datasets, our target settings were primarily single-task. Although the **Droid-Multitask** setting partially addresses this, extending evaluation to truly large-scale, multi-task robotics settings remains an open and impactful avenue. Addressing these limitations will be crucial for scaling DataMIL to practical robotic applications.

## 7    REPRODUCIBILITY STATEMENT

We have taken several steps to ensure the reproducibility of our results. Sec. 5 describes all experimental settings in detail. In addition, App. B provides step-by-step descriptions and pseudocode for estimating datamodels with both regression- and metagradient-based approaches, App. F outlines the full experimental setup for our simulation and real-world environments, along with the composition of the prior datasets, and App. H lists key hyperparameters we use in DataMIL for each setting. Together, these materials are intended to facilitate transparent replication of our findings.

### ACKNOWLEDGMENTS

We thank Luca Macesanu for helping with some of the real world evaluations and Bowen Jiang for his feedback on the manuscript. We also thank Jullian Yapeter and Karl Pertsch for their assistance in generating the MetaWorld dataset. Work supported in part by DARPA TIAMAT program (HR0011-24-9-0428) and also supported by the NSF grant DMS-2134108 and Open Philanthropy.

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

## A    QUALITATIVE RESULTS: WHAT DATA IS SELECTED BY DATAMIL?

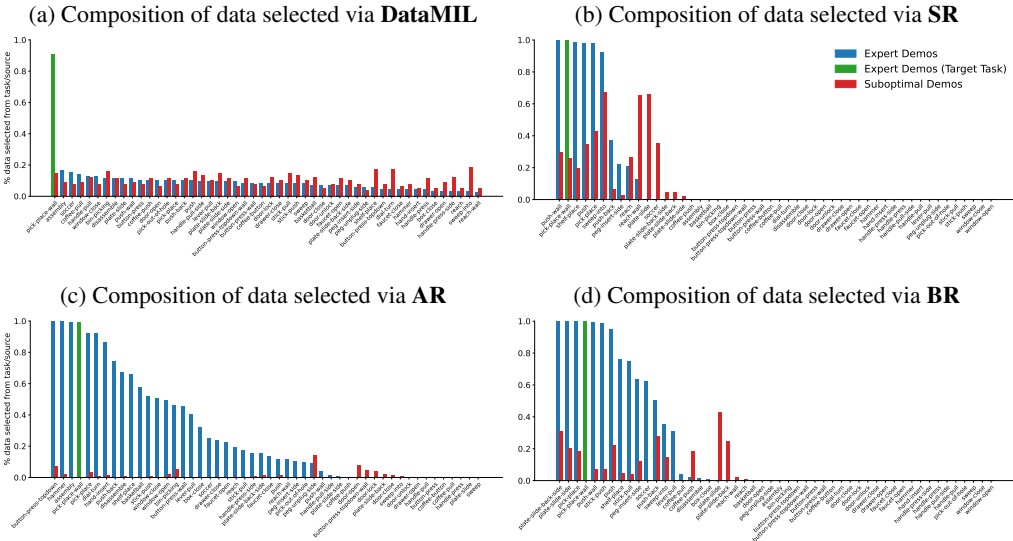

Figure 7: **MetaWorld Qualitative Results.** Percentage of data selected from each task and expert/suboptimal source for **DataMIL**, **SR**, **AR** and **BR**

### A.1    METAWORLD

As described in Appendix F.1, our prior dataset for MetaWorld combines both expert and suboptimal demonstrations. In our experiments, we retrieve the top 10% of this data ranked by DataMIL to train the policy. Here, we qualitatively examine the dataset selected by DataMIL for the **pick-place-wall** task.

Figure 7 shows the percentage of data selected from each task and data source (expert or suboptimal). We observe that **SR**, while able to retrieve samples from relevant tasks, fails to differentiate between expert and sub-optimal demonstrations—resulting in the inclusion of a large fraction of low-quality data. In contrast, **AR** filters out sub-optimal samples more effectively by matching actions, but it lacks task awareness due to its disregard for state information, often pulling data from irrelevant tasks. **BR**, which embeds both state and action features jointly, exhibits a blend of SR and AR behaviors—capturing elements of both but also inheriting their limitations. In comparison, **DataMIL** consistently selects data from the correct task (green bar) while also avoiding noisy, sub-optimal examples.

### A.2    LIBERO

In LIBERO, for two target tasks, Book-Caddy (Figure 8) and Bowl-Cabinet (Figure 9), we investigate the proportion of data that is selected from each of the prior tasks in LIBERO-90.

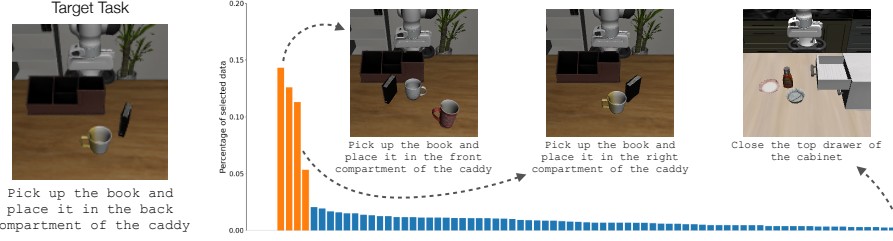

Figure 8: **LIBERO Bool-Caddy.** Proportion of prior tasks selected for the book-caddy task

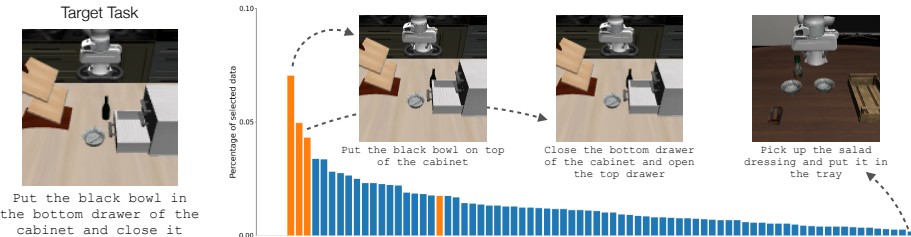

Figure 9: **LIBERO Bowl-Cabinet.** Proportion of prior tasks selected for the bowl-cabinet task

Since the exact target tasks do not appear in the prior dataset, we manually label (in orange) tasks with similar semantics and object layouts. We also visualize examples from the most and least frequently selected tasks. The plots support empirically that DataMIL consistently selects data from tasks with matching object and scene configuration. The visualizations also reflect this, showing that related tasks from similar looking scenes, albeit with related but different language instructions, are most frequently selected, aligning with intuition developed in prior works (Saxena et al., 2025; Maddukuri et al., 2025).

### A.3    OXE

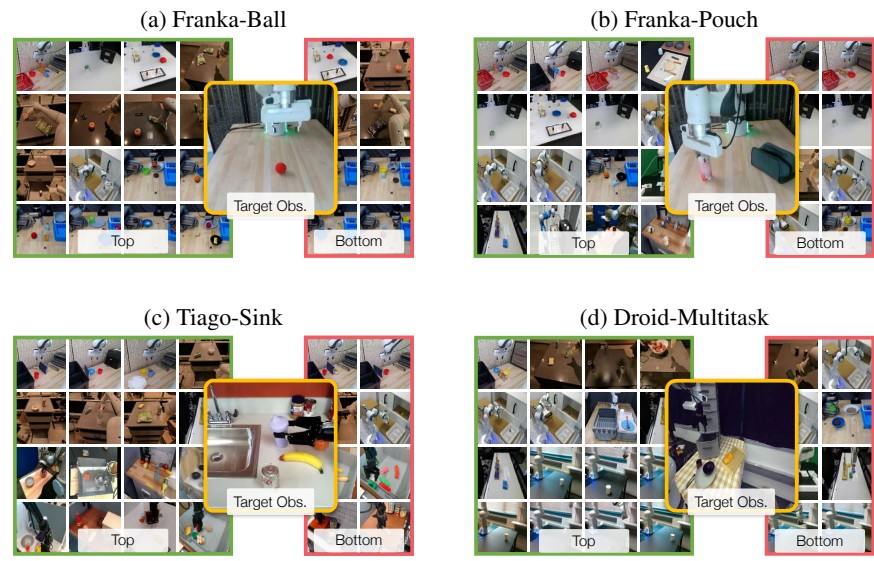

Figure 10: Top and bottom ranked samples by **DataMIL** for each of the real-world tasks

Figure 10 visualizes the highest and lowest ranked datapoints identified by DataMIL for real-world tasks. As discussed in the main text, these examples often appear visually similar, aligning with findings in computer vision (Ilyas et al., 2022; Feldman & Zhang, 2020) where data points that look alike but carry different labels can mislead the model. In robotics, this may occur when visually similar observations correspond to different actions, thereby confusing the policy. How and why these fine-grained differences affect data selection, and ultimately, the policy performance, is an interesting direction for future work.

## B    ESTIMATING DATAMODELS

In this section, we describe the datamodeling framework in more detail. In particular, we first provide the formal version of Informal Definition 1, then describe the estimators that we use to construct datamodels in this work, namely the regression estimator and the metagradient-based estimator.

## B.1 FORMALIZING DATAMODELING

The goal of datamodeling is to construct a function $\hat{f}$ that can predict the performance of a learning algorithm $\mathcal{A}$ on any given subset of the data $\mathcal{D}' \subset \mathcal{D}$ without actually training a policy on that subset. Let $\mathcal{D}$ be a prior dataset of imitation-learning data of size $N = |\mathcal{D}|$, and let us represent any subset of $\mathcal{D}$ as a binary vector $\mathbf{w} \in \{0, 1\}^N$ where $w_i = 1$ if the $i$-th training sample is in the subset and $w_i = 0$ otherwise. Let a learning algorithm $\mathcal{A}$ be a function that takes as input a dataset (represented as a binary vector $\mathbf{w}$) and outputs a policy $\mathcal{A}(\mathbf{w})$. The datamodeling problem is to construct a function $\hat{f}$ that can predict the performance of $\mathcal{A}(\mathbf{w})$ when trained on any given subset of the data without actually training a policy on that subset. More formally, we aim to find a function $\hat{f}$ minimizing the following loss:

$$\mathbb{E}_{\mathbf{w}\sim\text{Bernoulli}(\frac{1}{2})^N} \left[ \left( \mathcal{M}(\mathcal{A}(\mathbf{w})) - \hat{f}(\mathbf{w}) \right)^2 \right], \tag{6}$$

where $\mathcal{M}$ is the target metric and $\mathbf{w} \sim \text{Bernoulli}(\frac{1}{2})^N$ is a random binary vector of length $N$.

Recall from the main text that we are particularly interested in datamodels $\hat{f}$ that are additive in the training dataset—in terms of our formalization, we are interested in functions $\hat{f}$ of the form

$$\hat{f}(\mathbf{w}) = \mathbf{w}^\top \beta. \tag{7}$$

for some vector $\beta \in \mathbb{R}^N$. An *estimation method* for datamodels is thus just a method for finding a good estimate of the vector $\beta$. We refer the reader to Ilyas et al. (2022) for a more detailed discussion of datamodeling.

## B.2 REGRESSION ESTIMATOR

The regression estimator is a simple yet effective method for estimating the vector $\beta$ that treats datamodeling as a supervised learning problem. In particular, the regression estimator first samples a set of $m$ binary vectors $\mathbf{w}_1, \ldots, \mathbf{w}_m \sim \text{Bernoulli}(p)^N$ where $p$ is the probability of inclusion; for each of these binary vectors, it trains a policy $\mathcal{A}(\mathbf{w}_i)$ on the subset of the data indexed by $\mathbf{w}_i$, and evaluates its performance using the target metric $\mathcal{M}$. It then fits a linear model to the performance of these policies on the sampled binary vectors, i.e., it solves

$$\min_{\beta\in\mathbb{R}^N} \frac{1}{m} \sum_{i=1}^m \left( \mathcal{M}(\mathcal{A}(\mathbf{w}_i)) - \mathbf{w}_i^\top \beta \right)^2, \tag{8}$$

and uses the resulting vector $\hat{\beta}$ as the parameters of the datamodel $\hat{f}(\mathbf{w}) = \mathbf{w}^\top \hat{\beta}$. The cost of building this estimator is high, since it requires training a policy for each of the $m$ binary vectors, but Ilyas et al. (2022) shows that the estimator can be very accurate, and indeed identifies highly influential subsets of the prior dataset. Pseudocode for the regression estimator is provided in Algorithm 1.

## B.3 METAGRADIENT-BASED ESTIMATOR

The metagradient-based estimator is a more sophisticated method for estimating the vector $\beta$ that avoids the high cost of training a policy for each binary vector. Instead, the metagradient-based estimator operates by leveraging a classical statistical tool called the *influence function* (Hampel, 1974). Intuitively, the metagradient-based estimator proceeds as follows. First, instead of thinking of training datasets as *binary* vectors $\mathbf{w} \in \{0, 1\}^N$, we think of them as *real-valued* vectors $\mathbf{w} \in [0, 1]^N$, where each coordinate $w_i$ corresponds to the *importance weight* placed on the $i$-th training sample in the dataset. Concretely, if $w_i = 0$, then the $i$-th training sample is not used in the training set, and if $w_i = 1$, then the $i$-th training sample is used in the training set with full weight; if $0 < w_i < 1$, then the $i$-th training sample is used in the training set but its loss is scaled by $z_i$. Observe that this parameterization is equivalent to the binary parameterization for $w_i \in \{0, 1\}$, but gives us a continuous way to represent the training set.

Once we have this continuous parameterization, we can write the *first-order approximation* to $\mathcal{M}(\mathcal{A}(\mathbf{w}))$ as,

$$\mathcal{M}(\mathcal{A}(\mathbf{w})) \approx \mathcal{M}(\mathcal{A}(\mathbf{w}_0)) + \nabla\mathcal{M}(\mathcal{A}(\mathbf{w}_0))^\top (\mathbf{w} - \mathbf{w}_0), \tag{9}$$

---

**Algorithm 1** Regression Estimator

**Input:** Dataset $D$ of size $N$, policy learner $\mathcal{A}(\cdot)$, target metric $\mathcal{M}(\cdot)$
**Hyperparameters:** binary masks $m$, prob. $p$

▷ Generate training pairs for the surrogate
$T \leftarrow []$
**for** $i \leftarrow 1$ **to** $m$ **do**
    Sample $\mathbf{w}_i \sim \text{Bernoulli}(p)^N$
    Train policy on subset: $\pi_i \leftarrow \mathcal{A}(\mathbf{w}_i)$
    Evaluate performance: $y_i \leftarrow \mathcal{M}(\pi_i)$
    Append pair: $T \leftarrow T + [(\mathbf{w}_i, y_i)]$

▷ Fit linear datamodel using Eq. 8
$$\hat{\boldsymbol{\beta}} \leftarrow \arg \min_{\boldsymbol{\beta} \in \mathbb{R}^N} \frac{1}{m} \sum_{i=1}^{m} (y_i - \mathbf{w}_i^\top \boldsymbol{\beta})^2$$
**return** $\hat{\boldsymbol{\beta}}$

---

**Algorithm 2** Metagradient-based Estimator

**Input:** initial included data $\mathbf{c} = \mathbf{1}_n$, policy learner $\mathcal{A}(\cdot)$, target metric $\mathcal{M}(\cdot)$
**Hyperparameters:** step size $p$, number of steps $T$

▷ Gradient Storage
$G \leftarrow []$
▷ Metagradient descent loop
**for** $t \leftarrow 1$ **to** $T$ **do**
    $\mathbf{w} \leftarrow \mathbf{0}_n$
    ▷ Compute metagradients
    $\mathbf{g} \leftarrow \dfrac{\partial \mathcal{M}(\mathcal{A}_\mathbf{c}'(\mathbf{w}))}{\partial \mathbf{w}}$
    Sample $\mathbf{m} \sim \text{Bernoulli}(p)^N$
    Store $\mathbf{g}$: $G \leftarrow G + [\mathbf{g} \odot \mathbf{m}]$
    ▷ Update Counts
    $\mathbf{c} \leftarrow \mathbf{c} - \text{sign}(\mathbf{g}) \odot \mathbf{m}$
**return** $G$

---

where $\mathbf{w}_0$ is the vector of all ones. The gradient $\nabla \mathcal{M}(\mathcal{A}(\mathbf{w}_0))$ is known as the *influence function*, and gives us a *linear* approximation to the loss in Equation 6. That is, if we could compute the influence function exactly, we could use it as a datamodel directly, i.e.,

$$\hat{f}(\mathbf{w}) = \mathcal{M}(\mathcal{A}(\mathbf{w}_0)) + \nabla \mathcal{M}(\mathcal{A}(\mathbf{w}_0))^\top (\mathbf{w} - \mathbf{w}_0). \tag{10}$$

Traditionally, the influence function is notoriously hard to compute, and so prior work on data attribution has focused on approximating it (Koh & Liang, 2017; Park et al., 2023; Bae et al., 2024). However, recent work has shown how to compute it *exactly* and *efficiently* (Engstrom et al., 2025) and how to use this exact influence function as a datamodel estimator (Ilyas & Engstrom, 2025).

Observe that in order for the metagradient-based estimator to be valid, the function $\mathcal{M}(\mathcal{A}(\mathbf{w}))$ must be *differentiable* with respect to $\mathbf{w}$. For this to be satisfied, it is sufficient for (a) the target metric $\mathcal{M}$ to be differentiable with respect to the policy $\mathcal{A}(\mathbf{w})$, and (b) the policy $\mathcal{A}(\mathbf{w})$ to be trained via an iterative algorithm composed of elementary differentiable operations (which is almost all of the popular off-the-shelf learning algorithms).

**Gradient descent on training data.** To operationalize the metagradient-based estimator, we use the *Metagradient Descent* (MGD) algorithm (Engstrom et al., 2025), to compute influence scores and iteratively refine the dataset. The algorithm views data selection as an optimization problem over the *selected data counts* $\mathbf{c} \in \{0,1\}^N$, where $c_i = 1$ indicates inclusion of the $i$-th sample.

Given a count vector $\mathbf{c}$, let $\mathcal{A}_\mathbf{c}$ denote the policy obtained by training on the dataset defined by $\mathbf{c}$, and let $\mathcal{M}(\mathcal{A}_\mathbf{c})$ denote its evaluation on the target metric. MGD introduces a differentiable surrogate $\mathcal{A}_\mathbf{c}'(\mathbf{w})$, where $\mathbf{w} \in \mathbb{R}^N$ perturbs the per-sample weights during training only at a certain iteration $k$, while keeping the rest of the training as is. By construction, setting $\mathbf{w} = \mathbf{0}$ recovers the original training procedure, i.e., $\mathcal{A}_\mathbf{c}'(\mathbf{0}) = \mathcal{A}_\mathbf{c}$.

We can then compute the influence/metagradient as,

$$\mathbf{g} = \nabla_\mathbf{w} \mathcal{M}(\mathcal{A}_\mathbf{c}'(\mathbf{w}))\Big|_{\mathbf{w}=\mathbf{0}}, \tag{11}$$

which measures how an infinitesimal upweighting of each training example at the $k$th iteration would affect the downstream performance. The sign of $\mathbf{g}_i$ thus reveals whether increasing or decreasing the weight of sample $i$ would improve $\mathcal{M}$.

To update the dataset, MGD applies stochastic coordinate updates: a random mask $\mathbf{m} \sim \text{Bernoulli}(p)^N$ selects a subset of coordinates to update, and the counts are modified as

$$\mathbf{c} \leftarrow \mathbf{c} - \text{sign}(\mathbf{g}) \odot \mathbf{m}, \tag{12}$$

where $\odot$ denotes elementwise multiplication. This step is analogous to performing gradient descent over training samples with $p$ as the *learning rate*. Repeating this process iteratively over $T$ steps

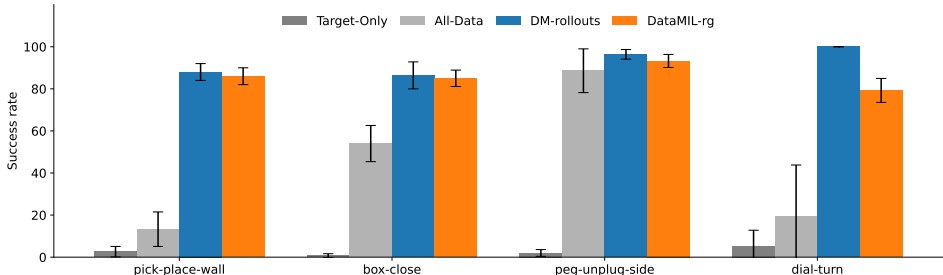

Figure 11: **Extended Fig. 2 Results.** Comparing true rollout success ($\mathcal{M}$) vs. proxy metric ($\widehat{\mathcal{M}}$)

refines **c**, yielding a dataset concentrated on the most influential examples. In our experiments, we found that averaging the gradients **g** computed over the $T$ iterations lead to more reliable influence scores. A pseudocode of the algorithm is provided in Algorithm 2 and we refer the reader to Engstrom et al. (2025) for more details.

## C   COMPARISONS AND ADDITIONAL RESULTS

Here we provide some additional results and show a comparison of some key hyperparameters such as percentage of data selected, cluster size and more.

### C.1   COMPARISON: TRUE ROLLOUT SUCCESS VS. PROXY METRIC

Here we provide extended results of our analysis in Figure 2. Recall from before, that we are comparing the data selected using datamodels computed via our proposed proxy metric in Equation 5 (*DataMIL-rg*) vs. true policy rollouts (*DM-rollouts*). We make this comparison in 4 tasks from MetaWorld as shown in Figure 11, where we select the top 10% of samples (as ranked by their estimated datamodel coefficients) from a multi-task prior dataset consisting of a mix of expert and suboptimal demonstrations (dataset details provided in Section F.1) and measure the success rates of policies trained on the selected data. Across all tasks we observe a similar trend as discussed in Section 4, (a) our proxy metric only incurs a small drop in performance across all tasks, thus demonstrating its efficacy as a viable metric for data selection and (b) data curation is critical: naively using all data or only target examples yields suboptimal policies, whereas curated datasets substantially boost task success.

### C.2   METAWORLD NO-GOAL CONDITIONING EXPERIMENTS

In Figure 3 of the main paper, we presented results on MetaWorld with *goal conditioning*, where policies receive explicit goal information provided by the simulator. Goal-conditioning is often essential in settings like LIBERO and OXE, where the target task has limited demonstrations and generalization from other tasks is required. However, in MetaWorld, the prior dataset already includes expert demonstrations for the target tasks. Therefore, an effective data selection method should be capable of retrieving relevant examples—even in the absence of goal information.

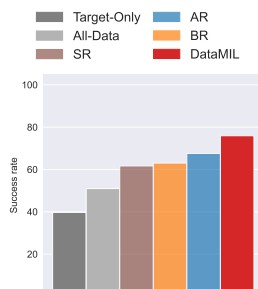

Figure 12: Avg. success on MetaWorld without goal conditioning.

To test this, we repeat the MetaWorld experiments described in Appendix F.1, but mask out goal states during both data selection and policy training. Results averaged over all 50 tasks are shown in Figure 12. Even without goal-conditioning, DataMIL continues to outperform the best baseline by 10% in average success rate. However, overall performance across all methods declines compared to the goal-conditioned setting (Figure 3a), highlighting that, without goal information, selected data from other tasks can introduce harmful interference during training.

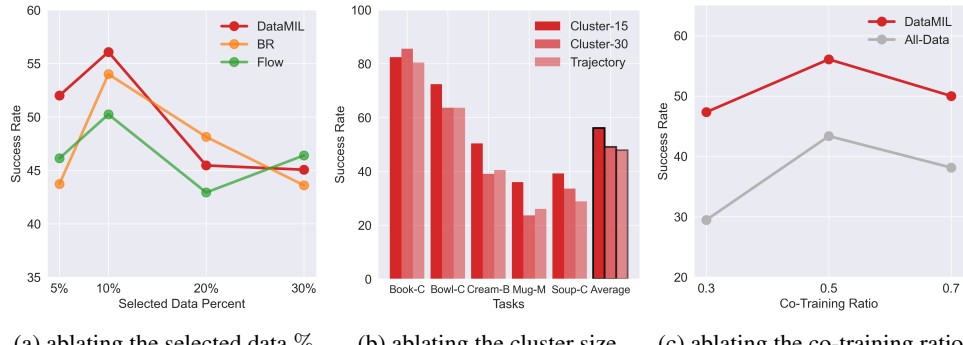

(a) ablating the selected data %   (b) ablating the cluster size   (c) ablating the co-training ratio

Figure 13: Ablation experiments of DataMIL over different key data selection and training hyperparameters. Each experiment is averaged over 5 seeds on each of the tasks in LIBERO-5. In (a) and (c) each data point represents an averaged success rate over all 5 tasks.

### C.3 COMPARISON: PERCENTAGE OF DATA SELECTED

A key hyperparameter in data selection is the proportion of prior data to select. Unfortunately, there is no principled way to determine this value other than training and evaluating policies across different selection ratios. To enable systematic ablations, we construct a smaller benchmark within the LIBERO suite, denoted LIBERO-5, consisting of five tasks: *Book-Caddy*, *Bowl-Cabinet*, *Cream-Butter*, *Mug-Microwave*, and *Soup-Cheese*.

Figure 13a reports the average performance of DataMIL and two baselines, *BR* (Du et al., 2023) and *Flow* (Lin et al., 2024), on LIBERO-5 while varying the percentage of prior data selected. At low selection ratios ($5\%$ and $10\%$), DataMIL substantially outperforms the baselines, highlighting its ability to prioritize higher-quality samples compared to heuristic-based methods. At higher selection ratios ($\geq 20\%$), the performance gap narrows and all methods achieve comparable results, as the majority of useful data is eventually included, diminishing the impact of the selection strategy.

### C.4 COMPARISON: CLUSTER SIZE

As described in Section 4.2, we group individual state–action pairs into temporal clusters before estimating their influence on policy performance via datamodels. We evaluate the effect of cluster granularity on LIBERO-5, with results shown in Figure 13b. Specifically, we compare three strategies: *Cluster-15* (sub-trajectories of length 15), *Cluster-30* (sub-trajectories of length 30), and *Trajectory* (entire trajectories as clusters). We observe that finer-grained clustering yields higher performance, as it provides more precise control over which segments of data are selected. Accordingly, our main experiments adopt a cluster size of 15.

### C.5 COMPARISON: CO-TRAINING RATIO

While the data selected by DataMIL can be leveraged in multiple ways, we adopt a co-training strategy that combines target data $\mathcal{D}_{target}$ and selected data $\mathcal{D}_{sel}$. At each training step, a batch is drawn from $\mathcal{D}_{target}$ with probability $\alpha$ and from $\mathcal{D}_{sel}$ with probability $1 - \alpha$. Although this choice is not intrinsic to DataMIL, it influences downstream performance. Figure 13c shows the average LIBERO-5 results of DataMIL and the *All-Data* baseline across different co-training ratios $\alpha$. We find that $\alpha = 0.5$ consistently yields the best results. Given the similar setup between our simulation and real-world experiments, and the high cost of real-world evaluations, we adopt $\alpha = 0.5$ for real-robot experiments as well.

### C.6 COMPARISON: LONGER TRAINING FOR ALL-DATA BASELINE

Since the *All-Data* baseline leverages substantially more training data than other methods, we also investigate whether it benefits from extended training. On LIBERO-5, we compare training durations of $1\times$, $2\times$, and $4\times$ the number of steps used for other methods. The resulting average

| Tasks | Target-Only | All-Data | AR | BR | Flow | STRAP | DataMIL (ours) |
|---|---|---|---|---|---|---|---|
| | | | **LIBERO Evaluations** | | | | |
| Soup-Sauce | $13.2 \pm 7.6$ | $20.0 \pm 15.6$ | $32.8 \pm 9.3$ | $38 \pm 15.3$ | $50 \pm 14.4$ | $33.2 \pm 15.10.7$ | $39.2 \pm 11.9$ |
| Cream-Butter | $27.6 \pm 6.5$ | $35.2 \pm 13.2$ | $39.6 \pm 10.7$ | $41.2 \pm 14.6$ | $47.2 \pm 10.8$ | $20.0 \pm 9.7$ | $50.4 \pm 8.6$ |
| Stove-Moka | $27.4 \pm 7.3$ | $24.0 \pm 9.8$ | $31.2 \pm 8.1$ | $30.4 \pm 10.0$ | $33.2 \pm 6.7$ | $43.6 \pm 5.5$ | $40.4 \pm 8.5$ |
| Bowl-Cabinet | $48.8 \pm 6.4$ | $65.6 \pm 8.3$ | $62.8 \pm 3.3$ | $73.6 \pm 8.5$ | $69.2 \pm 12.2$ | $77.2 \pm 10.3$ | $72.4 \pm 5.2$ |
| Mug-Mug | $0.4 \pm 0.9$ | $2.0 \pm 2.44$ | $4.8 \pm 2.3$ | $2.0 \pm 2.8$ | $6.4 \pm 6.5$ | $5.2 \pm 5.2$ | $0.8 \pm 1.1$ |
| Book-Caddy | $51.2 \pm 8.3$ | $58.4 \pm 10.7$ | $65.2 \pm 13.1$ | $76.8 \pm 7.6$ | $69.2 \pm 13.2$ | $83.2 \pm 7.8$ | $82.4 \pm 1.7$ |
| Mug-Pudding | $2.0 \pm 2.0$ | $2.4 \pm 1.7$ | $5.6 \pm 5.9$ | $8.4 \pm 2.2$ | $5.8 \pm 3.5$ | $10.8 \pm 6.9$ | $4.8 \pm 3.3$ |
| Soup-Cheese | $11.2 \pm 3.3$ | $22.0 \pm 6.5$ | $29.3 \pm 10.4$ | $35.2 \pm 5.2$ | $26.8 \pm 4.8$ | $35.2 \pm 3.9$ | $36.0 \pm 6.0$ |
| Moka-Moka | $5.6 \pm 4.3$ | $12 \pm 8.2$ | $4.8 \pm 2.3$ | $15.2 \pm 4.1$ | $7.6 \pm 5.5$ | $6.8 \pm 1.8$ | $12.0 \pm 5.5$ |
| Mug-Microwave | $27.6 \pm 13.5$ | $35.6 \pm 10.4$ | $29.2 \pm 10.6$ | $43.2 \pm 9.9$ | $38.8 \pm 5.2$ | $34.4 \pm 8.3$ | $39.2 \pm 12.7$ |
| **Libero-Average** | 21.5 | 27.72 | 30.52 | 36.4 | 35.42 | 34.96 | **37.76** |
| | | | **OXE Evaluations** | | | | |
| **Franka-Ball** | 21.4 | 35.7 | 28.6 | 50.0 | 28.6 | 57.1 | 50.0 |
| **Franka-Pouch** | 17.6 | 17.6 | 11.8 | 41.2 | 35.3 | 29.4 | 70.6 |
| **Tiago-Sink** | 33.3 | 43.6 | 35.9 | 46.2 | 46.2 | 38.5 | 64.1 |
| Droid (Drawer) | 0.0 | 0.0 | 0.0 | 20.0 | 70.0 | 55.0 | 75.0 |
| Droid (Bread) | 4.2 | 33.3 | 20.8 | 0.0 | 41.7 | 16.7 | 41.7 |
| Droid (Napkin) | 25.0 | 45.0 | 40.0 | 45.0 | 40.0 | 35.0 | 65.0 |
| **Droid-Multitask** | 9.4 | 26.6 | 20.3 | 20.3 | 50.0 | 34.4 | 59.4 |
| **Real-Average** | 20.4 | 30.9 | 24.1 | 39.4 | 40.0 | 39.8 | **61.0** |

Table 1: Numerical results for LIBERO and OXE evaluations

performances are 43.36, 44.13, and 38.40, respectively. These results indicate no consistent benefit from longer training; in fact, excessive training can degrade performance due to overfitting. For fairness, we therefore use the same number of training steps across DataMIL and all baselines.

## C.7    NUMERICAL RESULTS AND DISCUSSION

Task-wise numerical success rates for the LIBERO and real-world settings are reported in Table 1. Our results on LIBERO differ from those reported in the original **STRAP** paper on a similar setting, and we attribute these differences to two key factors:

1. **Evaluation Protocol.** We suspect that the main contributing factor is likely a difference in the evaluation protocol. In the original STRAP implementation, the authors evaluate multiple training checkpoints and report results from the best-performing model. This can lead to higher reported success rates. In contrast, our evaluation protocol follows a stricter setup: we evaluate the policy only once, at the final checkpoint after training completes, without any checkpoint selection.

2. **Policy Architecture.** While we use the original STRAP data retrieval code, we differ in the policy architecture used for imitation learning. Specifically, we train with Octo (Octo Model Team et al., 2024), a large transformer-based diffusion policy, whereas STRAP uses a transformer-based policy from Robomimic (Mandlekar et al., 2021) with a Gaussian mixture head. These two models have different inductive biases, which can lead to variation in performance across tasks. For example, when trained solely on the five target demonstrations from the *moka-moka* task, Octo achieves a 6% success rate, while Robomimic achieves 0%. However, in the *mug-mug* task, the Robomimic policy reaches 38% success, while Octo performs close to 0%.

We believe that both protocol differences and model architecture contribute to the gap in reported numbers, and since our goal is to study data selection, our results reflect a fair and consistent evaluation under a unified training and assessment setup across all baselines.

# D PROXY METRIC DETAILS

Recall from Eq. 5 that our general proxy metric is

$$\widehat{\mathcal{M}}(\pi, \mathcal{D}_{target}) = \frac{1}{|\mathcal{D}_{target}|} \sum_{(s,a) \in \mathcal{D}_{target}} -\mathcal{L}_{BC}(\pi(s), a)$$

where $\mathcal{L}_{BC}$ is the behavior-cloning loss appropriate to the policy class. Here we provide how the equation looks like for specific policy classes that we used in our experiments.

**MetaWorld.** We parametarize the policy in MetaWorld as,

$$\pi_\theta(a \mid s) = \mathcal{N}\big(a; \mu_{\theta_1}(s), \, \text{diag}(\sigma_{\theta_2}(s)^2)\big)$$

and consider two forms of $\mathcal{L}_{BC}$:

1. *Negative Log-Likelihood (NLL)*:

$$\begin{aligned}
\mathcal{L}_{\text{NLL}}(s, a) &= -\log \pi_\theta(a \mid s) \\
&= \tfrac{1}{2}\left(a - \mu_{\theta_1}(s)\right)^\top \Sigma(s)^{-1}\left(a - \mu_{\theta_1}(s)\right) \;+\; \tfrac{1}{2}\log\det\big(2\pi\,\Sigma(s)\big)
\end{aligned} \tag{13}$$

2. *$l_1$ loss*:

$$\mathcal{L}_{\ell_1}(s, a) \;=\; \big\|a \;-\; \mu_{\theta_1}(s)\big\|_1 \tag{14}$$

Empirically, the $l1$ loss works better for the *regression estimator*, while *metagradient-based estimator* performs well with the NLL loss.

**LIBERO and OXE.** In the LIBERO and OXE settings, we use Octo as the learning model which is a transformer-based policy with a diffusion action head. It's behavior-cloning loss is the standard denoising score-matching objective:

$$\mathcal{L}_{\text{diff}}(s, a) = \mathbb{E}_{t \sim \text{Uniform}[1,T], \, \epsilon \sim \mathcal{N}(0,I)} \left\| \epsilon - \epsilon_\theta\big(\sqrt{\bar{\alpha}_t}\, a + \sqrt{1 - \bar{\alpha}_t}\, \epsilon, \; s, \; t\big) \right\|^2$$

where $\alpha_t \in (0, 1)$ is the forward-process noise schedule, $\bar{\alpha}_t = \prod_{i=1}^t \alpha_i$, and $\epsilon_\theta(a_t, s, t)$ is the network's noise prediction.

Substituting $\mathcal{L}_{BC} = \mathcal{L}_{\text{diff}}$ into the proxy metric gives,

$$\widehat{\mathcal{M}}(\pi, \mathcal{D}_{\text{target}}) \;=\; -\frac{1}{|\mathcal{D}_{\text{target}}|} \sum_{(s,a) \in \mathcal{D}_{\text{target}}} \mathbb{E}_{t,\epsilon} \left\| \epsilon - \epsilon_\theta\big(\sqrt{\bar{\alpha}_t}\, a + \sqrt{1 - \bar{\alpha}_t}\, \epsilon, \; s, \; t\big) \right\|^2 \tag{15}$$

## D.1 UP-WEIGHTING RELEVANT STATES

In robotics, more often than not we have some information about what states and actions are of higher importance than others, for example states closer to object interactions may be more relevant than moving around in free space. Our proxy metric provides a seamless way to incorporate prior knowledge by re-weighting important states in the behavioral cloning loss. Concretely, we introduce a state-action-dependent weight $w(s, a)$ into the objective:

$$\widehat{\mathcal{M}}(\pi, \mathcal{D}_{target}) = \frac{1}{|\mathcal{D}_{target}|} \sum_{(s,a) \in \mathcal{D}_{target}} -\underline{w(s,a)}\mathcal{L}_{BC}(\pi(s), a)$$

By default, we set $w(s, a) = 1$ and use the unweighted proxy. In the LIBERO experiments, however, we found that doubling the weight for states immediately preceding a grasp significantly improves data selection. Thus, for those "pre-grasp" states we use $w(s, a) = 2$, while all other states retain $w(s, a) = 1$.

## E  BASELINE IMPLEMENTATION

Implementation details of the baselines are provided below.

- **FlowRetrieval** (Lin et al., 2024) and **BehaviorRetrieval** (Du et al., 2023): FlowRe-trieval (**Flow**) and Behavior Retrieval (**BR**) baselines compute similarity on the image flows and state-action pairs respectively. Since these features typically include high-dimensional image observations, **Flow** and **BR** train VAEs to encode the features into a more manage-able latent space, which they can use to compute similarities between prior and target data. We used the implementation provided by the authors of FlowRetrieval (Lin et al., 2024) for training the VAEs and computing the similarity for both, **Flow** and **BR**, in the LIBERO and OXE settings (`https://github.com/lihenglin/bridge_training_code`).

  For both **Flow** and **BR** (and other heuristics such as Action Retrieval (**AR**) and State Re-trieval (**SR**)), once each state in the prior data is assigned a score based on the similarity measure, we select the top $x\%$ of the data most similar to the target where $x$ is the selection budget.

- **STRAP** (Memmel et al., 2024): In the LIBERO experiments, we use the authors' **STRAP** implementation ( `https://github.com/WEIRDLabUW/STRAP` ) to embed sub-trajectories with DinoV2 (Oquab et al., 2023) and compute similarity via dynamic time warping. **STRAP** expects HDF5-formatted inputs, but our OXE pipeline relies on TFDS. We therefore adapted the **STRAP** code to accept TFDS datasets without altering its core logic.

  While **STRAP**'s original recommendation is to retrieve the top 100 sub-trajectories in LIBERO, we found that training Octo on these segments underperforms. Instead, we retrieve the most similar sub-trajectories until they constitute 10% of the prior data—matching the budget used by DataMIL and our other baselines. In LIBERO, this modification boosts success from **24.72%** (with 100 segments) to **34.96%** (our reported results) averaged over LIBERO-10 tasks. We apply the same retrieval strategy on OXE, sampling sub-trajectories until we match the selection size of our method and baselines.

For MetaWorld, we found it more effective to compute similarity over temporal windows rather than individual state–action pairs. Specifically, for each baseline (**BR**, **SR**, **AR**), we slide a fixed-length horizon $\mathcal{H}$ over both prior and target data, concatenate each segment's states (and actions) into a single high-dimensional vector, and then measure similarity between these flattened vectors. This horizon-based approach captures temporal context, enabling the baselines to reject noisy or suboptimal samples—ultimately improving retrieval quality and downstream policy performance. Our initial experiments found $\mathcal{H} = 50$ to perform best.

## F  TRAINING AND EVALUATION DETAILS

### F.1  METAWORLD

**Dataset.** The MetaWorld dataset is constructed from two sources: scripted expert policies and reinforcement learning (RL) exploration. MetaWorld provides scripted policies for each of its 50 tasks, which we use to generate 4,000 episodes totaling 350K environment steps. For the RL data, we train a multi-task SAC agent on all 50 tasks for 12 million transitions, reaching an average success rate of 21%. To create a representative prior dataset, we uniformly subsample from the SAC replay buffer across all tasks, yielding 1 million environment steps—approximately 3× larger than the scripted data. For each target task, we generate 5 expert demonstrations using the scripted policy as $\mathcal{D}_{target}$.

**Datamodel estimation using DataMIL.** We cluster the prior dataset at the trajectory level and use 5 demonstrations from $\mathcal{D}_{target}$ to compute the proxy objective. We then compute the datamodels using the *regression-based* datamodel estimator. The top 10% of prior trajectories, ranked by the datamodels, are selected to form $\mathcal{D}_{sel}$.

**Policy Training and Evaluation.** We train a behavior cloning policy with an MLP backbone and a tanh-squashed Gaussian output distribution from garage (garage contributors, 2019) on $\mathcal{D}_{sel}$. In

Table 2: Datasets utilized across the OXE subsets in our experimental setup

| Dataset | OXE13 | OXE23 | OXE24 | Dataset | OXE13 | OXE23 | OXE24 |
|---|---|---|---|---|---|---|---|
| RT1 | ✓ | ✓ | ✓ | CMU IAM Lab | ✓ | ✓ | ✓ |
| Viola | ✓ | ✓ | ✓ | Roboturk | ✗ | ✓ | ✓ |
| Austin Buds | ✓ | ✓ | ✓ | BC-Z | ✗ | ✓ | ✓ |
| Austin Mutex | ✓ | ✓ | ✓ | CMU Stretch | ✗ | ✓ | ✓ |
| Austin Sailor | ✓ | ✓ | ✓ | DLR Edan | ✗ | ✓ | ✓ |
| Austin Sirius | ✓ | ✓ | ✓ | Berkeley Autolab UR5 | ✗ | ✓ | ✓ |
| Taco Play | ✓ | ✓ | ✓ | Berkeley Fanuc | ✗ | ✓ | ✓ |
| Jaco Play | ✓ | ✓ | ✓ | Berkeley Cable | ✗ | ✓ | ✓ |
| Stanford Hydra | ✓ | ✓ | ✓ | Bridge | ✗ | ✓ | ✓ |
| NYU Franka | ✓ | ✓ | ✓ | NYU Door Opening | ✗ | ✓ | ✓ |
| Furniture Bench | ✓ | ✓ | ✓ | Toto | ✗ | ✓ | ✓ |
| UCSD Kitchen | ✓ | ✓ | ✓ | Kuka | ✗ | ✗ | ✓ |

preliminary experiments, we found that co-training with $\mathcal{D}_{target}$ yielded negligible improvements, so we exclude it in this setting. Each policy is trained and evaluated over 3 random seeds.

### F.2  LIBERO

**Dataset.** Our prior dataset consists of 4,500 human teleoperated demonstrations from LIBERO-90, with 50 demonstrations per task. The 10 tasks from LIBERO-10 serve as our target tasks. For each, we randomly sample 5 demonstrations to form the target dataset $\mathcal{D}_{target}$.

**Datamodel estimation using DataMIL.** Prior to running DataMIL, we segment the prior demonstrations into sub-trajectories of horizon length 15, which we found to provide a good balance between granularity and noise-robustness. We then estimate influence scores using the *metagradient-based estimator* with the weighted proxy metric described in Appendix D.1. The top 10% of sub-trajectories, based on datamodel influence, are selected to form the selected dataset $\mathcal{D}_{sel}$.

**Policy Training and Evaluation.** We fine-tune a language-conditioned Octo policy, starting from the publicly released Octo-small checkpoint, by co-training on $\mathcal{D}_{target}$ and $\mathcal{D}_{sel}$ using a co-training ratio $\alpha = 0.5$ for 10k steps. We evaluate each policy on the corresponding target task using 50 rollouts and report results averaged over 5 random seeds.

### F.3  OXE

Table 3: Task-wise experimental setup for selecting data and, training and evaluating policies

| | Embodiment | Prior Dataset | Prior Selection Ratio | No. of Target Demos | No. of Evaluations |
|---|---|---|---|---|---|
| Franka-Pick | Franka-Panda | OXE13 | 1% | 10 | 14 |
| Franka-Pouch | Franka-Panda | OXE23 | 0.75% | 30 | 17 |
| Tiago-Sink | Tiago | OXE24 | 0.5% | 20 | 39 |
| Droid-Multitask | Franka-Panda | OXE24 | 1% | 40 (total) | 32 (total) |
| Drawer | - | - | - | 10 | 10 |
| Bread | - | - | - | 15 | 12 |
| Napkin | - | - | - | 15 | 10 |

**Dataset.** We use subsets of the Open X-Embodiment (OXE) dataset (O'Neill et al., 2024) as our prior data. Specifically, we define three subsets—**OXE13**, **OXE23**, and **OXE24**—with their respective constituent datasets listed in Table 2. The mapping between tasks and dataset subsets is shown in Table 3. For each task, we collect a separate target dataset via teleoperation (Dass et al., 2024), varying the number of demonstrations per task based on difficulty (see Table 3).

**Datamodel estimation using DataMIL.** Following the DataMIL recipe, we cluster the prior data at the trajectory level and estimate influence using the metagradient-based estimator with our proposed proxy objective. To reduce distribution shift during datamodel training, we split $\mathcal{D}_{target}$ into two halves: one half is used to compute the proxy metric, while the other is included in the training mix. After training, we select the top $x\%$ of prior trajectories based on influence scores, where $x$ is specified per task in Table 3.

**Policy Training and Evaluation.** We fine-tune a language-conditioned Octo-small checkpoint using co-training on $\mathcal{D}_{target}$ and the selected dataset $\mathcal{D}_{sel}$, with a co-training ratio $\alpha = 0.5$ for 50k steps. Final policy is evaluated on the corresponding target tasks using a fixed number of real-world rollouts (Table 3). To ensure fair comparisons, we fix the spatial configurations of relevant objects across all methods. For instance, in the **Franka-Pouch** task, we use 17 predefined object poses (position and orientation) for evaluation. In the more challenging **Droid-Multitask** setting, we report both full and partial successes as part of the final success rate (e.g., a partially closed drawer or grasping the bread/napkin), with partial completions weighted as 0.5. All real-world evaluations were conducted using a single seed.

## G  COMPUTATION COSTS FOR DATA SELECTION

We report the computational costs associated with both estimators, measured across MetaWorld, LIBERO, and OXE.

**Regression estimator.** The regression estimator requires training and evaluating a policy on $m$ random subsets of the dataset (Alg. 1). While increasing $m$ improves estimation quality, the computational cost is dominated by policy training, which scales linearly with $m$. In our MetaWorld experiments, we found $m = 10^4$ sufficient to obtain high-quality datamodels. This setting required a total of 8 GPU hours on $8\times$A100 GPUs (equivalent to roughly 60–70 GPU hours on a single GPU). Although this may appear expensive for a single task, the same $m$ trained policies can be reused to compute the proxy objective across all 50 tasks in the MetaWorld suite, effectively amortizing the datamodel training cost to $\sim$1–2 hours per task.

**Metagradient-based estimator.** The metagradient-based estimator is typically 3–5$\times$ slower than standard policy training, due to the additional cost of computing metagradients. Taking multiple steps (hyperparameter $T$ in Alg. 2) further improves estimation quality but incurring a higher cost. In MetaWorld, data selection completes within $\sim$1 hour on a single A5000 GPU. On LIBERO, parallel training across 4 A5000s requires 16–20 hours—comparable to the training times of VAE-based BR (Du et al., 2023) and Flow (Lin et al., 2024) baselines. For OXE, which is substantially larger (300k trajectories), training takes 3–4 days on 4 A100 GPUs. Notably, DataMIL avoids online rollouts, making training passive and able to run without any human oversight.

## H  HYPERPARAMETERS FOR ESTIMATING DATAMODELS

Below we provide the hyperparameters used for estimating the datamodels via regression and metagradient-based methods (refer to Alg. 1 and 2). For experiments related details (e.g. amount of data selected), please refer to App. F.

**Regression estimator.** We utilize the regression estimator only in the MetaWorld setting. Specifically, we construct training pairs by sampling datapoints with probability $p = 0.1$, training a $400 \times 3$ MLP policy with ReLU activations and a tanh-squashed Gaussian action head, and evaluating its loss on a held-out dataset. This procedure is repeated $m = 10^4$ times to generate the training set used to regress the datamodel parameters.

**Metagradient-based estimator.** The hyperparameters used for the metagradient-based estimator for each of the settings is listed in Table 4. When we compute the metagradients, instead of computing it over the entire training process (which would be too expensive), we compute it over only a segment of training. Hence, the hyperparameter $k$ refers to the number of steps at the end of training, over which the metagradients are computed.

Table 4: Hyperparameters for the Metagradient-based Estimator (Alg. 2).

| Hyperparameter | Metaworld | LIBERO | OXE |
|---|---|---|---|
| step size $p$ | 1.0 | 1.0 | 0.125 |
| number of steps $T$ | 30 | 30 | 20 |
| policy architecture | gaussian MLP | Octo-Small | Octo-Small |
| policy learner $\mathcal{A}(\cdot)$ | | | |
|    training steps | 1100 | 2100 | 5100 |
|    metagradient steps $k$ | 100 | 100 | 100 |
|    optimizer | adamw | adam | adam |
|    lr schedular | 1cycle | cosine | cosine |

## I  LLM USAGE

We complied with the ICLR 2026 policies on LLM usage by using LLMs only to polish the writing and improve clarity — correcting grammar, improving phrasing, and enhancing readability. We did not use LLMs for generating original content, producing research ideas, or analyzing data. Any substantive scientific content are the result of our own work, and we take full responsibility for them.

