# OpenReview forum: "DataMIL: Selecting Data for Robot Imitation Learning with Datamodels"
_ICLR.cc/2026/Conference — ICLR 2026 Poster_

### Official Review · Reviewer_FrUR · 2025-10-16

**Soundness:** 4
**Presentation:** 4
**Contribution:** 3
**Rating:** 8
**Confidence:** 4

**Summary:**

This paper introduces DataMIL, a framework for selecting the most beneficial data from large, heterogeneous prior datasets to improve imitation learning for specific robotic tasks. The core problem addressed is that naively co-training with large datasets can harm performance, while heuristic-based selection methods (e.g., based on visual or semantic similarity) are often suboptimal because they don't directly consider the data's impact on the final policy's performance.


DataMIL adapts the "datamodels" paradigm to robotics. It works by estimating the influence of each data point (or cluster of data points) on a policy's task success. To make this feasible for robotics, it introduces two key adaptations:


1. A surrogate objective: Instead of using expensive real-world rollouts to measure performance during data selection, DataMIL uses the policy's validation loss on a small set of target-task data as a differentiable proxy metric.

2. Efficient estimators: It explores two methods for calculating data influence scores: a comprehensive but computationally expensive Regression Estimator and a more efficient Metagradient-based Estimator suitable for large, modern policies like Octo.


The proposed pipeline involves clustering prior data, estimating influence scores for each cluster using the surrogate objective, selecting the top-ranked clusters, and finally co-training a policy on this selected data and the target task data. The authors validate DataMIL across 60+ simulation and real-world tasks, demonstrating superior performance over several baselines by selecting data from large datasets like MetaWorld, LIBERO-90, and the Open X-Embodiment (OXE) dataset.

**Strengths:**

- As robotic datasets like OXE and DROID grow in scale and heterogeneity, the problem of "negative transfer" or performance degradation from irrelevant data becomes increasingly severe. DataMIL provides a concrete solution to this important and timely challenge in the field.
- The paper's primary strength is its shift from heuristic-based data selection to a more principled, end-to-end optimization. By directly estimating a data point's influence on policy performance (via a proxy), DataMIL moves beyond potentially misleading similarity metrics and focuses on the ultimate goal: task success. This is a significant conceptual improvement over methods that rely purely on visual, motion, or state-action similarity.
- The paper successfully adapts a powerful idea from the broader machine learning community to the specific constraints of robotics. The introduction of a differentiable, rollout-free surrogate loss is a clever and necessary innovation that makes the datamodels framework tractable for physical systems.
- The authors provide comprehensive validation across a wide range of settings: a large number of simulation tasks in MetaWorld (50) and LIBERO (10), and challenging real-world tasks using the massive and heterogeneous OXE dataset. The demonstrated ability to select useful data for an unseen embodiment (Tiago robot) is particularly compelling.

**Weaknesses:**

- The entire method hinges on the assumption that minimizing validation loss on a small target dataset ($\hat{\mathcal{M}}$) is a reliable proxy for maximizing real-world success rate ($\mathcal{M}$). The paper provides one experiment (Fig. 2) to show this correlation holds, but this is on a single task. This proxy is known to be noisy in robotics. It is plausible that for tasks involving complex contact dynamics, deceptive local minima, or sparse rewards, data that merely helps "fit" the target demonstrations (i.e., lower BC loss) might not be the data that promotes robust, generalizable behavior. The proxy might favor data that is visually simple or has similar action patterns, even if other data could teach a more valuable physical prior, potentially filtering out more diverse or challenging data from the prior dataset that could have been crucial for improving robustness and generalization.
- The method introduces several critical hyperparameters that significantly impact performance, and the paper relies on empirical ablations to set them.
1. Figure 10a shows that performance is sensitive to the percentage of data selected. There is no principled way offered to determine the optimal ratio a priori, meaning users must perform a costly sweep.

2. Figure 10b demonstrates that performance varies with cluster granularity. The justification that the optimal strategy is a "function of the dataset size" is intuitive but not actionable without further guidance.

3. The paper fixes α=0.5 based on simulation results (Fig. 10c), but this optimal ratio is likely task- and dataset-dependent. The need for these expensive, nested hyperparameter sweeps undermines the overall efficiency of the framework.

- The cross-embodiment selection for the Tiago robot (Fig. 5a) is interesting, but the paper only observes that the selected data represents "robots operating on a table top from an ego-perspective". This is a very general description. It's unclear if DataMIL is discovering abstract skills or simply matching on coarse visual scene structure, the latter being a much weaker form of transfer.

**Questions:**

While the validation loss proxy appears to work well in your experiments, its reliability is a critical assumption. Can you characterize the potential failure modes? In what specific types of robotic tasks (e.g., those with high physical variability, tasks requiring tool use where initial states are similar but require vastly different actions, or long-horizon tasks with sparse success signals) might minimizing validation loss lead to the selection of data that is detrimental to real-world task success?

---

> ### Author Response · Authors · 2025-11-25
> **Response to reviewer (1/1)**
>
> **Q. What are the potential failure modes of the validation-loss proxy? For which classes of robotic tasks might minimizing validation loss cause DataMIL to select data that ultimately harms real-world success?**
>
> A. We empirically validate the reliability of our proxy objective in Fig. 2, and, following the reviewer’s suggestion, we extended this analysis to several additional tasks in MetaWorld (a visualization can be found [here](https://imitation-datamodel.github.io/datamil.github.io/rebuttal_figures/mw_proxy_extended.pdf)),
>
> | Method        | pick-place-wall | box-close     | peg-unplug-side | dial-turn     |
> |---------------|------------------|---------------|------------------|----------------|
> | Target-Only   | 2.6 ± 2.5        | 0.7 ± 0.94     | 2.0 ± 1.6        | 5.3 ± 7.5      |
> | All-Data      | 13.3 ± 8.2       | 54.0 ± 8.6    | 88.6 ± 10.4      | 19.3 ± 24.5    |
> | DM-rollouts   | 88.0 ± 4.0       | 86.4 ± 6.4    | 96.4 ± 2.3       | 100.0 ± 0.0    |
> | DataMIL-rg    | 86.0 ± 4.0       | 85.0 ± 3.9    | 93.25 ± 3.1      | 79.25 ± 5.7    |
>
> The results follow the trend in Fig.2 and indicate that the data selected using our proposed proxy metric tracks the true metric well in most tasks, only incurring a small drop in final policy success rate.
>
> That said, we acknowledge that the proxy may fail in certain settings. Potential failure modes may include,
> 1. **Highly precise or fine-grained control tasks:** When correct and incorrect action distributions differ only subtly, their behavioral cloning losses can be nearly indistinguishable, making it difficult for the proxy to rank demonstrations effectively.
> 2. **Tasks where certain states matter disproportionately:** Because the proxy averages loss uniformly across the target trajectory, it may miss demonstrations that are only critical in a few key states (e.g., contact initiation, tool alignment). We discuss a simple extension using state-action weighting in Appendix D.1 when domain knowledge is available, though we agree that developing automatic weighting strategies is an important direction for future work.
>
> Overall, while our proxy performs well across the tasks we tested, we recognize that tasks with subtle action differences or highly uneven state importance may challenge its reliability.
>
> ### Further points of clarification in the review
> > The method introduces several critical hyperparameters that significantly impact performance…
>
> While DataMIL introduces several hyperparameters, many of them, such as the data-selection percentage and co-training ratio, are not unique to our method and are also required by existing data selection and retrieval baselines. We agree that reducing sensitivity to these hyperparameters is an important direction for future work. More broadly, as with many new methodologies, we expect that hyperparameter practices will become more standardized and intuitive as the approach matures and is adopted by the community.
>
> > …It's unclear if DataMIL is discovering abstract skills or simply matching on coarse visual scene structure, the latter being a much weaker form of transfer.
>
> Our results in Fig. 4 suggest that DataMIL captures more than coarse visual similarity. Baseline methods such as STRAP and BR—which rely primarily on visual matching—perform substantially worse across all tasks, indicating that DataMIL is leveraging richer task-relevant structure. That said, we agree that developing better tools for interpreting the selected data would provide deeper insight into what abstract skills or priors are being transferred, and this remains an interesting direction for future work.

---

### Official Review · Reviewer_FHqk · 2025-10-30

**Soundness:** 3
**Presentation:** 3
**Contribution:** 3
**Rating:** 6
**Confidence:** 3

**Summary:**

The paper proposed a data selection strategy called DataMIL for robot imitation learning problems. DataMIL is built upon prior work (datamodels) on data selection in CV and NLP problems: the regression method and the metagradient method. To tackle the intractable datamodeling objective in robot imitation learning, the paper proposed some modifications to the standard datamodels: (1) a proxy metric, (2) clustering demonstrations temporally, and (3) a strategy to prevent distribution shift. Empirically, the DataMIL achieved higher success rates on both simulated and real-world multi-task robotics benchmarks.

**Strengths:**

- Instead of using heuristics such as visual similarity or state–action closeness to select subsets of prior data, this paper proposed to use the performance-aware datamodels to predict scores for selecting each data, resulting in an end-to-end data selection algorithm for robotics imitation learning problem.

- The proposed proxy metric prevents expensive and intractable rollouts on real-robot during data selection and also prevents repeatedly learning the same algorithms on different subsets of the prior data.

- Empirically, the evaluations of the proposed methods range from both simulated and real-world robotics benchmarks and show consistent gains of DataMIL over multiple baselines.

**Weaknesses:**

- Gradient-based data reweighting and data balancing for supervised learning problems has been widely studied in the literature [1, 2, 3]. The concepts of datamodels and regression based methods have been proposed in prior methods. Also, the data selection for robotics imitation learning is still a supervised learning problem. Overall, the contributions of the paper seem to be incremental.

- The explanations for some strategies in Sec. 4.2 are unclear and some ablation experiments are limited. See the question part for details.

- Since the paper focuses on empirical studies, including preliminary open-source code in the submission would strengthen the conclusions.

[1] Ren, Mengye, Wenyuan Zeng, Bin Yang, and Raquel Urtasun. "Learning to reweight examples for robust deep learning." In International conference on machine learning, pp. 4334-4343. PMLR, 2018.

[2] Shu, Jun, Qi Xie, Lixuan Yi, Qian Zhao, Sanping Zhou, Zongben Xu, and Deyu Meng. "Meta-weight-net: Learning an explicit mapping for sample weighting." Advances in neural information processing systems 32 (2019).

[3] Wang, X., Pham, H., Michel, P., Anastasopoulos, A., Carbonell, J., & Neubig, G. (2020, November). Optimizing data usage via differentiable rewards. In International Conference on Machine Learning (pp. 9983-9995). PMLR.

**Questions:**

- Fig 2: Is the target-only policy trained on a dataset with expert and sub-optimal demonstrations on the target task? I am curious about the performance of imitation learning only on target expert demonstrations.

- The ablation experiments for the proxy metric $\hat{M}$ in Sec 4.2 only limits to a single task.

- The “clustering training examples” paragraph in Sec. 4.2: The description of the specific clustering methods are not clear from the texts. Is there any ablation study showing the effects and necessity of clustering training examples?

- Fig 2, Fig 3, Fig 4, and Sec 5.2: Is there a direct quantitative comparison between the dataset selected by different methods? For example, if imitation learning on the target expert data can achieve almost 100% success rate, then computing the IoU score or some clustering based scores (e.g., NMI score) between these expert data and data selected by different methods would be direct comparisons.

- Any ideas about how to perform data selection only on fine-tuning datasets and how to reduce the distribution shift if we cannot retain the prior dataset?

- Any comments on computing the metric as in Eq.5 if the policy objective itself is not differentiable, e.g., if we were doing RL instead of IL.

---

> ### Author Response · Authors · 2025-11-25
> **Response to reviewer (1/2)**
>
> **Q. Is the target-only policy in Fig.2 trained on a dataset with expert and sub-optimal demonstrations on the target task?**
>
> A. In Fig. 2, the target-only policy is trained only on the 5 expert demonstrations for the target task; no sub-optimal data is included. The small amount of target data leads to the policy’s low performance.
>
> **Q. The ablation experiments for the proxy metric $\hat{M}$ in Sec. 4.2 only limits to a single task.**
>
> A. We thank the reviewer for highlighting this. To address this we have extended the analysis in Fig.2 to several other tasks in MetaWorld and provide the results in the table below (a visualization can be found [here](https://imitation-datamodel.github.io/datamil.github.io/rebuttal_figures/mw_proxy_extended.pdf)),
>
> | Method        | pick-place-wall | box-close     | peg-unplug-side | dial-turn     |
> |---------------|------------------|---------------|------------------|----------------|
> | Target-Only   | 2.6 ± 2.5        | 0.7 ± 0.94     | 2.0 ± 1.6        | 5.3 ± 7.5      |
> | All-Data      | 13.3 ± 8.2       | 54.0 ± 8.6    | 88.6 ± 10.4      | 19.3 ± 24.5    |
> | DM-rollouts   | 88.0 ± 4.0       | 86.4 ± 6.4    | 96.4 ± 2.3       | 100.0 ± 0.0    |
> | DataMIL-rg    | 86.0 ± 4.0       | 85.0 ± 3.9    | 93.25 ± 3.1      | 79.25 ± 5.7    |
>
> The results follow the trend in Fig.2 and indicate that the data selected using our proposed proxy metric tracks the true metric well in most tasks, only incurring a small drop in final policy performance. We will add these results in the revised version of our manuscript.
>
> **Q. The description of the specific clustering methods are not clear from the texts. Is there any ablation study showing the effects and necessity of clustering training examples?**
>
> A. Yes, Appendix C.3 includes an ablation comparing different clustering sizes in LIBERO. Here we clarify the clustering strategy from Sec. 4.2: in principle, computing influence at the level of individual samples would offer the most flexibility. However, in large datasets many samples are seen only a few times (or sometimes not at all) during training, which leads to very noisy influence estimates. Averaging influence over nearby state-action pairs helps reduce this noise, which motivates computing influence at the cluster level.
>
> A simple heuristic we use is to choose a cluster size that ensures each cluster is encountered several times during training. Robotics data naturally has strong temporal consistency and we leverage it by forming clusters as either full trajectories or fixed-length sub-trajectories. For sub-trajectory clustering, we set a fixed horizon length and split each trajectory into chunks of that length, though other clustering strategies (including variable-length sub-trajectories) are also compatible with DataMIL.
>
> **Q. Is there a direct quantitative comparison between the dataset selected by different methods? For example, if imitation learning on the target expert data can achieve almost 100% success rate, then computing the IoU score or some clustering based scores (e.g., NMI score) between these expert data and data selected by different methods would be direct comparisons.**
>
> A. We appreciate the suggestion, and we performed a related analysis in Appendix A.1 for the MetaWorld pick-place-wall task. There, we report the fraction of data selected from each task and show that DataMIL reliably selects the relevant expert demonstrations while filtering out suboptimal demos and irrelevant tasks.
>
> However, quantitative overlap metrics such as IoU or NMI have two practical limitations in this setting. First, there is rarely a well-defined “ground truth” or golden set of optimal demonstrations in prior datasets; in the MetaWorld example, we identified such a set heuristically using domain knowledge, but this does not generalize across tasks or environments. Second, the optimal dataset is not necessarily unique and different subsets may lead to equally strong policies. In such cases, overlap-based metrics could unfairly penalize methods that select different but equally useful data. For these reasons, we measure usefulness in the most direct and objective way: evaluating how well a policy trained on each selected dataset performs (Sec. 5.1). This directly measures the utility of the selected data for the intended downstream use case.

---

> ### Author Response · Authors · 2025-11-25
> **Response to reviewer (2/2)**
>
> **Q. Any ideas about how to perform data selection only on fine-tuning datasets and how to reduce the distribution shift if we cannot retain the prior dataset?**
>
> A. If data selection must be performed only on the fine-tuning dataset (with no access to the pre-training data), our procedure remains unchanged. We can fine-tune the model for a few iterations on this new dataset, compute a datamodel using the metagradient-based estimator on these examples, and then curate the fine-tuning data based on the attribution scores. DataMIL does not require access to the original pre-training dataset in this stage.
>
> For the second part of the question on “reducing the distribution shift if we cannot retain the prior dataset,” we are not fully sure which scenario the reviewer has in mind. We would be very grateful if the reviewer could clarify this point so that we can provide a more precise answer.
>
> **Q. Any comments on computing the metric as in Eq.5 if the policy objective itself is not differentiable, e.g., if we were doing RL instead of IL.**
>
> A. This is an important question, and several approaches could make Eq. 5 workable when the policy objective is non-differentiable, as in RL:
> 1. **Use the policy gradient trick.** Replace the target metric with the return 𝐽(𝜋) and use policy gradients to obtain a differentiable surrogate. Concurrent work such as [1] explores this idea in an online, single-task RL setting. This would also require costly online rollouts to compute.
> 2. **Use a learned critic.** Replace the target metric with a differentiable value estimator, such as a Q-function trained from offline or online rollouts. Gradients can then be propagated through the critic, similar to DDPG-style methods. This is a promising direction for future work.
>
> > ...preliminary open-source code in the submission would strengthen the conclusions.
>
> We agree that open-source code is valuable for transparency and reproducibility. We have ensured we provide all details, summarized in Section 7, to reproduce the results of the paper. Further, we are committed to open-science and will release the full codebase upon acceptance.
>
> [1] CUPID: Curating Data your Robot Loves with Influence Functions, Agia et al., 2025

---

> > ### Comment · Reviewer_FHqk · 2025-11-26
> >
> > > reducing the distribution shift if we cannot retain the prior dataset
> >
> > This question asks for the scenario where we cannot retain the prior dataset for co-training, e.g., the pre-training and fine-tuning paradigm, and the downstream task has a distribution shift from the pre-training datasets. In general, using imitation learning to do pre-training can result in sub-optimal initialization (worse than random initialization) when fine-tuned on an out-of-distribution task.
> >
> > I thank the authors for their detailed response. I am generally satisfied with the responses and would like to maintain my positive evaluation.
> >
> > At the same time, I suggest that the authors incorporate new experiments and include the following discussions in the paper:
> > - data selection for learning paradigms other than co-training.
> > - how to compute the metric as in Eq.5 if the policy objective itself is not differentiable.

---

> > > ### Author Response · Authors · 2025-12-03
> > > **Follow-up response to reviewer (1/1)**
> > >
> > > We thank the reviewer for the helpful clarification and are glad that our responses addressed the main concerns.
> > >
> > > Regarding the clarified scenario, where the prior dataset cannot be retained, and pre-training may produce a suboptimal initialization for out-of-distribution downstream tasks. As noted in our earlier response, DataMIL can operate directly on the fine-tuning dataset: we fine-tune the model briefly on the new data, compute influence scores, and curate the fine-tuning dataset. The data curation itself helps mitigate the distribution shift since it allows to correct for potential misalignment by identifying the fine-tuning examples that most improve the downstream objective on the target domain. Co-training on the selected data as well as the target domain data, further helps to reduce this shift. We agree that exploring other ways of minimizing distribution shift is a promising direction for future work, and we appreciate the reviewer highlighting this point. Finally, our MetaWorld experiments use randomly initialized policies, whereas LIBERO and OXE experiments use a pretrained policy. We show that in both settings, DataMIL selects demonstrations that meaningfully improves downstream performance.
> > >
> > > We also thank the reviewer for the constructive suggestions on strengthening the paper. We will revise the manuscript to include the additional experiments we performed above and provide a broader discussion on,
> > > 1) Data selection for learning paradigms other than co-training like the pre-training and finetuning paradigm,
> > > 2) Handling non-differentiable objectives such as in RL, covering how we can leverage policy-gradient surrogates or learned critics in Eq.5.
> > >
> > > We appreciate the reviewer's positive evaluation and their thoughtful guidance on improving the clarity and scope of the paper.

---

### Official Review · Reviewer_PBcc · 2025-11-01

**Soundness:** 4
**Presentation:** 3
**Contribution:** 4
**Rating:** 8
**Confidence:** 3

**Summary:**

This paper tackles the critical problem of selecting beneficial data from large, heterogeneous datasets for robot imitation learning. The authors argue that standard heuristic-based selection methods (e.g., visual or state-action similarity) are suboptimal and can even harm performance. To address this, the paper proposes DataMIL, a framework based on the 'datamodels' paradigm, which aims to predict a policy's performance based on the data it is trained on. The core contribution is a surrogate objective—the behavior cloning (BC) loss on a small set of target-task data. This surrogate avoids the need for expensive and risky real-world rollouts and is fully differentiable, enabling efficient metagradient-based estimation. The conclusion is supported by comprehensive experiments in simulation (MetaWorld, LIBERO) and on real-world hardware (OXE), demonstrating superior performance over several baselines.

**Strengths:**

- The proposed problem is critical for the robot learning community. As the field increasingly relies on scaling up data (e.g., OXE), methods for effectively curating this data to specialize in new tasks are essential.
- The paper's core idea of using a differentiable, rollout-free surrogate metric (BC loss on target data) is novel and effective. It elegantly adapts the datamodels framework to the specific constraints of robotics.
- The evaluation is comprehensive, spanning multiple benchmarks (MetaWorld, LIBERO, OXE), policy architectures (MLP, Octo), and strong heuristic baselines (BR, Flow, STRAP).
- The real-world evaluation on the heterogeneous OXE dataset is a significant strong point, especially the demonstration of successful cross-embodiment data selection (for the Tiago robot), which strongly supports the claims.

**Weaknesses:**

- The proposed data model incurs high computational costs, as it requires per-task estimation.
- The method is far from "plug-and-play," as it introduces numerous new hyperparameters that demand meticulous tuning, often relying on human heuristics and empirical expertise.

**Questions:**

- Q1: Is there a trade-off strategy to efficiently identify the approach that yields the maximal performance improvement for the policy?
- Q2: The authors find that the optimal clustering strategy differs between LIBERO (sub-trajectories) and OXE (full-trajectories). Could the authors provide a principled heuristic for selecting the appropriate clustering granularity based on dataset characteristics (e.g., dataset size, degree of heterogeneity, number of tasks)?
- Q3: I'm curious about the potential impact of leveraging the identified "negative samples" explicitly during training, rather than only using positive examples. For instance, could a contrastive learning objective be designed to actively repel these negative examples while attracting positive ones? Intuitively, explicitly instructing the policy "what not to learn" might be more efficient or precise for defining the correct decision boundaries than only providing positive examples.

---

> ### Author Response · Authors · 2025-11-25
> **Response to reviewer (1/1)**
>
> **Q. Is there a trade-off strategy to efficiently identify the approach that yields the maximal performance improvement for the policy?**
>
> A. The choice between the regression and metagradient variants depends primarily on model-training cost. The regression estimator is most effective when many models can be trained and evaluated quickly on random data subsets; it works well for small models with highly optimized training pipelines where each run completes in minutes. However, in most practical robotics settings (e.g. involving images) training even a single model is slow, making the regression approach prohibitively expensive. In these cases, the metagradient-based estimator is the preferred option: it is dramatically faster, and in our experiments delivers performance comparable to the regression-based method.
>
> **Q. Can a heuristic for choosing clustering granularity (sub-trajectory vs full-trajectory) be devised based on dataset characteristics such as size, heterogeneity, and number of tasks?**
>
> A. Our experiments suggest that the right clustering granularity depends on three factors: dataset size, compute budget, and the length of each trajectory. We provide some insights below,
>
> **Size of dataset and compute budget:** If the dataset is small or compute is not a constraint, using fine-grained clusters (individual samples or short sub-trajectories) works best because the method gets more flexibility in choosing what to keep or discard. For example, in LIBERO (~4500 demos), we use sub-trajectories of length 15; since each sample is seen multiple times, the influence estimates are stable (an ablation of different clustering sizes is provided in Appendix C.3). For large datasets, each sample may be seen only once or twice (and some samples may never be seen), making per-sample influence estimates noisy. In these cases, using coarser clusters (e.g., full trajectories or larger chunks) helps reduce the noise in the estimates. A good heuristic is to choose a cluster size that ensures each cluster appears often enough during datamodel training.
>
> **Length of trajectories:** If trajectories are very long and contain several sub-tasks, averaging influence over the entire trajectory can wash out important differences: some parts may be helpful while others may be harmful. In such cases, smaller clusters help DataMIL pick out only the useful segments.
>
> **Q. Could negative samples identified by DataMIL be exploited more explicitly, e.g., using contrastive objectives that push the policy away from harmful demonstrations instead of simply discarding them?**
>
> A. In this work we focused on identifying and using positive samples to improve policy performance, but we agree that explicitly leveraging negative samples is a very exciting direction. Negative examples could help define unsafe regions, guide/constrain exploration and more. Exploring how best to integrate these signals, whether through contrastive losses, or reinforcement-style reweighting, is an interesting extension that we leave for future work.
>
> ### Further points of clarification in the review
>
> > The proposed data model incurs high computational costs, as it requires per-task estimation.
>
> We agree that training datamodels introduce additional compute overhead compared to standard fine-tuning or simple heuristics, especially for very large policies. However, our design choices substantially mitigate this cost. First, we discuss two methods for estimating datamodels, of which the metagradient-based estimator is significantly (approximately 50x) more efficient compared to the regression estimator. Second, in practice we do not need to train each policy to convergence: a single pass over the dataset is sufficient to obtain reliable influence estimates, reducing per-policy training time by roughly 5-10x in our LIBERO and OXE experiments.
>
> Finally, recent work [1] shows that datamodels learned with smaller, lightweight models can accurately approximate the data attribution patterns of larger models, and can be a promising alternative when scaling up these techniques: select data using smaller models that are fast to train, and then train larger models on the curated dataset.
>
> > The method is far from "plug-and-play," as it introduces numerous new hyperparameters...
>
> While DataMIL introduces several hyperparameters, many of them, such as the data-selection percentage and co-training ratio, are not unique to our method and are also required by existing data selection and retrieval baselines. We agree that reducing sensitivity to these hyperparameters is an important direction for future work. More broadly, as with many new methodologies, we expect that hyperparameter practices will become more standardized and intuitive as the approach matures and is adopted by the community.
>
> [1] Small-to-Large Generalization: Data Influences Models Consistently Across Scale, Khaddaj et al., 2025

---

### Official Review · Reviewer_DUws · 2025-11-05

**Soundness:** 2
**Presentation:** 4
**Contribution:** 3
**Rating:** 6
**Confidence:** 4

**Summary:**

The paper tackles the challenge of improving task-specific robot imitation learning using large, diverse prior datasets that often contain helpful and harmful examples. Motivated by the limits of heuristic data filtering, which may select visually or semantically similar but suboptimal samples, the authors introduce DataMIL, a data-selection framework built on datamodels that estimates how each datapoint influences policy performance end-to-end. Their method avoids costly real-world rollouts by using a differentiable surrogate loss and meta-gradient or regression-based estimators to assign influence scores, selecting the most beneficial data while discarding harmful samples. Experiments across simulated and real manipulation tasks, including MetaWorld, LIBERO, and OXE, show consistent improvements over prior state-of-the-art retrieval and co-training baselines, particularly in heterogeneous real-world settings. The key takeaway is that performance-aware, end-to-end data attribution enables better curation of large robot datasets and significantly improves fine-tuning outcomes on new tasks.

**Strengths:**

1. The paper builds on the datamodels paradigm and adapts it thoughtfully to robotics, incorporating principled techniques like metagradients and surrogate losses to avoid expensive rollouts.
2. The paper demonstrates consistent improvements across 60+ simulation and real-world tasks (MetaWorld, LIBERO, OXE), making the empirical evidence strong and diverse.
3. Comparisons against multiple state-of-the-art retrieval and co-training methods, plus ablations on surrogate loss vs true success metrics, meaningfully validate the approach.
4. Demonstrates utility in real robot settings and across embodiments, an important step toward practical general-purpose robot learning.

**Weaknesses:**

1. Even with meta-gradients, building datamodels requires multiple policy trainings, making the pipeline significantly more expensive than standard fine-tuning or heuristic retrieval. This may limit usability in real-world labs without large compute.
2. The method is evaluated primarily on one new task at a time. Broader deployment to large-scale continual or many-task adaptation scenarios remains underexplored.
3. The core datamodel assumption is borrowed from NLP/CV; a deeper theoretical justification for robotic sequential decision-making settings would strengthen the contribution.
4. I was deeply interested in looking at more qualitative data selection examples by DataMIL and considered baselines. While the authors provide some examples, I did not find them to be very convincing. It would be nice to see more examples along the lines of Figure 6 for Meta World. Moreover, retrieved demo examples shown for real-world tasks (Figure 8) also do not tell me much - why did DataMIL select those demos? It would be nice to include more discussion around qualitative results in the paper.
5. While the paper compared against several retrieval methods in robotics datasets, simpler retrieval strategies, such as retrieving dataset with matching camera poses, objects being manipulated, etc., were not considered. Without such baselines, I believe it is hard to justify this method as the current best for retrieval-based learning from large-scale robotics datasets.


Other minor issues:
1. Caption in Figure 6 should be updated to show which task the plots are for.

**Questions:**

1. Did the authors analyze the outputs of DataMIL datamodel against some of the recent heuristic-based retrieval strategies for co-training? E.g. [1] and [2] have said that matching data distributions around specific factors such as camera poses and placement arrangements helps boost policy learning. According to [3], not all factors are equally important. Do those takeaways still hold for the datamodels trained in this paper?
2. In line 408, the authors say that they “replace the All-Data baseline with a Random baseline” due to the scale of OXE. What exactly is the issue due to the scale of OXE? Does the policy underfit the entire data? In which case, what if the policy architecture is scaled up - is retrieval still necessary?


References:

[1] What Matters in Learning from Large-Scale Datasets for Robot Manipulation, Saxena et al., 2025

[2] Sim-and-Real Co-Training: A Simple Recipe for Vision-Based Robotic Manipulation, Maddukuri et al., 2025

[3] Decomposing the Generalization Gap in Imitation Learning for Visual Robotic Manipulation, Xie et al., 2023

---

> ### Author Response · Authors · 2025-11-25
> **Response to reviewer (1/2)**
>
> **Q. How does DataMIL’s selected data compare against heuristic retrieval strategies based on camera poses and object placements as explored in [1, 2]? Do the factor-based takeaways hold from [3]?**
>
> A. We appreciate the reviewer’s question. Heuristic strategies based on camera poses or object placements are indeed insightful, but they typically require additional environment-specific information, such as calibrated camera extrinsics or precise object identities and positions. Such metadata is often unavailable in large-scale robot datasets; for example, OXE does not include camera angle annotations or object-level information, making a direct comparison infeasible. While we can’t directly compare these parameters, our qualitative analysis in Section 5.2 suggests that DataMIL typically selects data from similar camera angles, for example, in the Tiago-Sink task with ego-centric observations, DataMIL predominantly selects ego-centric demonstrations as well, consistent with the takeaway from [3] that imitation-learned policies are sensitive to the camera-pose factor.
>
> Following the reviewer’s suggestion, we extended our qualitative study to LIBERO tasks and examined how the selected data relates to scene layout. For two target tasks, Book-Caddy and Bowl-Cabinet, the figures below show the proportion of selected data coming from each prior task,
> - Book-Caddy figure can be found [here](https://imitation-datamodel.github.io/datamil.github.io/rebuttal_figures/libero_qualitative_book_caddy.pdf).
> - Bowl-Cabinet figure can be found [here](https://imitation-datamodel.github.io/datamil.github.io/rebuttal_figures/libero_qualitative_bowl_cabinet.pdf).
>
> Since the exact target tasks do not appear in the prior dataset, we manually label (in orange) tasks with similar semantics and object layouts. We also visualize examples from the most and least frequently selected tasks. The plots support empirically that DataMIL consistently selects data from tasks with matching object and scene configuration. The visualizations also reflect this, showing that related tasks from similar looking scenes, albeit with related but different language instructions, are most frequently selected, aligning with intuition developed in prior works [1, 2]. We will add this analysis in Appendix A in the revised version of the paper.
>
> **Q. Why was the All-Data baseline replaced with Random on OXE? Does the policy underfit in the All-Data setting, and if so, would increasing model capacity eliminate the need for retrieval?**
>
> A. Training the All-Data baseline on OXE is substantially more expensive than training on a random subset, and our preliminary experiments on the Franka-Ball task showed that both achieve similar performance. Given the large cost of training the All-Data baseline and the lack of observed benefit from using all data, we opted to report the Random baseline in our experiments.
>
> While we did not explicitly test model underfitting on OXE, our results in LIBERO show that a larger model capacity does not eliminate the need for retrieval. In LIBERO, where the All-Data baseline is trained on a much smaller dataset (~4500 demos) compared to OXE, its performance is still significantly worse than when using curated, task-relevant data (see Fig. 3). We also increased the compute budget for the All-Data baseline (Appendix C.5) and observed no notable performance improvement. These findings indicate that the core issue is not model size or training time, but the presence of heterogeneous or irrelevant demonstrations, and that performance benefits arise from selecting the right data rather than simply scaling capacity.
>
>
> [1] What Matters in Learning from Large-Scale Datasets for Robot Manipulation, Saxena et al., 2025.
> [2] Sim-and-Real Co-Training: A Simple Recipe for Vision-Based Robotic Manipulation, Maddukuri et al., 2025.
> [3] Decomposing the Generalization Gap in Imitation Learning for Visual Robotic Manipulation, Xie et al., 2023.

---

> ### Author Response · Authors · 2025-11-25
> **Response to reviewer (2/2)**
>
> ### Further points of clarification in the review
> >  ...building datamodels requires multiple policy trainings...This may limit usability in real-world labs without large compute.
>
> We agree that training datamodels introduce additional compute overhead compared to standard model training/fine-tuning, especially for large policies. However, in practice we do not need to train each policy to convergence: a single pass over the dataset is sufficient to obtain reliable influence estimates, reducing per-policy training time by roughly 5-10x in our LIBERO and OXE experiments. Additionally, recent work [1] shows that datamodels learned with smaller, lightweight models can accurately approximate the data attribution patterns of larger models, and can be a promising alternative when scaling up these techniques: select data using smaller models that are fast to train, and then train larger models on the curated dataset.
>
> > The method is evaluated primarily on one new task at a time. Broader deployment to large-scale continual or many-task adaptation scenarios remains underexplored.
>
> While our primary focus is single-task adaptation, we did evaluate DataMIL in a multi–target-task setting through the DROID-Multitask experiments, where the target set included three distinct tasks (bread-in-bowl, napkin-in-drawer, open-drawer). In this setting, DataMIL successfully selected data that improved performance across all tasks (see Fig. 4 and Table 1), demonstrating that the method can extend beyond the single-task regime. We agree that further investigation of data selection to broader continual or many-task adaptation scenarios is an exciting direction for future work and could further strengthen the applicability of our approach.
>
> > ....It would be nice to include more discussion around qualitative results in the paper.
>
> We have provided some more qualitative analysis from the LIBERO benchmark above and will be adding it to the revised version of the paper.
>
> [1] Small-to-Large Generalization: Data Influences Models Consistently Across Scale, Khaddaj et al., 2025

---

### Author Response · Authors · 2025-12-03
**Summary of the Rebuttal**

We thank the reviewers for providing constructive feedback on our work. For the AC’s convenience, below we summarize key points from the rebuttal. We have addressed all the questions from the reviewers and performed 2 additional experiments to strengthen our claims in the paper. We will be incorporating this feedback in the final revision.

 **Some reviewers (FHqk, FrUR) pointed out that our analysis in Figure 2 was only limited to a single task.**
We acknowledged this limitation and addressed it by extending the analysis in Fig.2 to several other tasks and provide the results in the table below (a visualization can be found [here](https://imitation-datamodel.github.io/datamil.github.io/rebuttal_figures/mw_proxy_extended.pdf)),

| Method        | pick-place-wall | box-close     | peg-unplug-side | dial-turn     |
|---------------|------------------|---------------|------------------|----------------|
| Target-Only   | 2.6 ± 2.5        | 0.7 ± 0.94     | 2.0 ± 1.6        | 5.3 ± 7.5      |
| All-Data      | 13.3 ± 8.2       | 54.0 ± 8.6    | 88.6 ± 10.4      | 19.3 ± 24.5    |
| DM-rollouts   | 88.0 ± 4.0       | 86.4 ± 6.4    | 96.4 ± 2.3       | 100.0 ± 0.0    |
| DataMIL-rg    | 86.0 ± 4.0       | 85.0 ± 3.9    | 93.25 ± 3.1      | 79.25 ± 5.7    |

The results follow the trend in Fig.2 and indicate that the data selected using our proposed proxy metric tracks the true metric well in most tasks, only incurring a small drop in final policy performance. We will include these results in the revision.

**Reviewers (DUws, FrUR) found our qualitative analysis of the selected data interesting and were seeking more insights.**
We extended our qualitative study to LIBERO and examined how the selected data relates to scene layout. For two target tasks, Book-Caddy and Bowl-Cabinet, the figures below show the proportion of selected data coming from each prior task. Because the exact target tasks do not appear in the prior dataset, we manually label (in orange) tasks with similar semantics and object layouts. We also visualize examples from the most and least frequently selected tasks.
- Book-Caddy figure can be found [here](https://imitation-datamodel.github.io/datamil.github.io/rebuttal_figures/libero_qualitative_book_caddy.pdf).
- Bowl-Cabinet figure can be found [here](https://imitation-datamodel.github.io/datamil.github.io/rebuttal_figures/libero_qualitative_bowl_cabinet.pdf).

The plots support empirically that DataMIL consistently selects data from tasks with matching object and scene configuration. We will include this analysis in Appendix A.

**Some reviewers (PBcc, FrUR) raised concerns regarding the use of hyperparameters, raising the tuning cost of our proposed method.**
While DataMIL utilizes several hyperparameters, we clarified to the reviewers that many of them, such as the data-selection percentage and co-training ratio, are not unique to our method and are also required by existing data selection and retrieval baselines. We agree that reducing sensitivity to these hyperparameters is an important direction for future work. More broadly, as with many new methodologies, we expect that hyperparameter practices will become more standardized and intuitive as the approach matures and is adopted by the community.

**Reviewers (DUws, PBcc) also highlighted that the compute cost of the method may be a limiting factor for broad adoption of the method.**
We clarified that while training datamodels introduces additional compute overhead compared to standard model training/fine-tuning, in practice, policies do not need to be trained to convergence: a single pass over the dataset is sufficient to obtain reliable influence estimates, reducing per-policy training time by roughly 5-10x in our LIBERO and OXE experiments. Additionally, recent work [1] shows that datamodels learned with smaller, lightweight models can accurately approximate the data attribution patterns of larger models, and can be a promising alternative when scaling up these techniques: select data using smaller models that are fast to train, and then train larger models on the curated dataset. This is an exciting avenue for future work.

**Discussion about performance metrics when they are non-differentiable, as in RL.**
We also thank the reviewer FHqk for prompting an interesting discussion about how to extend DataMIL when the performance metric of the policy may not be straightforwardly differentiable as in RL. We outlined two potential approaches using– 1) the policy-gradient surrogates or 2) trained differentiable critics, and will add this discussion in Section 6 of our paper.

[1] Small-to-Large Generalization: Data Influences Models Consistently Across Scale, Khaddaj et al., 2025

---

### Meta-Review · Area_Chair_hjr4 · 2026-01-06

**Summary:**

The reviewers' primary concerns that are still outstanding after rebuttal include:
* Overhead: Building datamodels makes the process significantly more expensive and slower than standard fine-tuning or heuristic retrieval. While it results in a superior final meal, it consumes far more time and energy.
* Hyperparameter Sensitivity: The method relies on factors like clustering granularity, selection percentages, and co-training ratios that lack principled guidance and require costly empirical tuning.
* Using BC validation loss as a proxy metric: There were significant doubts about whether validation BC loss is a robust proxy for real-world success.
* Some reviewers argued that adapting existing datamodel frameworks from other fields represented an incremental contribution.

I think the benefits outweigh the concerns. On the other hand, it is indeed an incremental work. Thus I recommend accepting as a poster.

**Reviewer Concerns:**

Addressed:
* Evaluation limited to single-task (Sec. 4.2): The authors addressed this concern by extending this analysis to four additional MetaWorld tasks, demonstrating that DataMIL consistently tracks the final rollout success rate with only a small performance drop.
* Needing broader deployment to large-scale continual tasks: The authors extended the qualitative study to LIBERO and demonstrated positive results.

Outstanding:
* Incremental Contribution
* Overhead
* Hyperparameter Sensitivity
* (Partially addressed) Using BC validation loss as a proxy metric: The authors empirically show that the proxy metric correlates well with the final results well. However, they also acknowledge that the proxy can fail in certain settings. Potential failure modes may include highly precise or fine-grained control tasks and tasks where certain states matter disproportionately.

**Reviewer Scores:**

My prediction
* DUws would have maintained a 6 as there was no major concern.
* PBcc would have maintained a 8 as there was no major concern.
* FHqk would have kept the 6 as they indicated so after rebuttal.
* FrUR may increase the score by 1 as questions about evaluation are answered.

---

### Decision · Program_Chairs · 2026-01-26

Accept (Poster)